# SAFE: Slow and Fast Parameter-Efficient Tuning for Continual Learning with Pre-Trained Models

**Linglan Zhao**[†,*]    **Xuerui Zhang**[§,*]    **Ke Yan**[†,✉]    **Shouhong Ding**[†]    **Weiran Huang**[‡,✉]

[‡] MIFA Lab, Qing Yuan Research Institute, SEIEE, Shanghai Jiao Tong University
[†] Tencent YouTu Lab    [§] Zhejiang University
{linglanzhao, kerwinyan, ericshding}@tencent.com
xrzhang0121@zju.edu.cn, weiran.huang@outlook.com

## Abstract

Continual learning aims to incrementally acquire new concepts in data streams while resisting forgetting previous knowledge. With the rise of powerful pre-trained models (PTMs), there is a growing interest in training incremental learning systems using these foundation models, rather than learning from scratch. Existing works often view PTMs as a strong initial point and directly apply parameter-efficient tuning (PET) in the first session for adapting to downstream tasks. In the following sessions, most methods freeze model parameters for tackling forgetting issues. However, applying PET directly to downstream data cannot fully explore the inherent knowledge in PTMs. Additionally, freezing the parameters in incremental sessions hinders models' plasticity to novel concepts not covered in the first session. To solve the above issues, we propose a Slow And Fast parameter-Efficient tuning (SAFE) framework. In particular, to inherit general knowledge from foundation models, we include a transfer loss function by measuring the correlation between the PTM and the PET-applied model. After calibrating in the first session, the slow efficient tuning parameters can capture more informative features, improving generalization to incoming classes. Moreover, to further incorporate novel concepts, we strike a balance between stability and plasticity by fixing slow efficient tuning parameters and continuously updating the fast ones. Specifically, a cross-classification loss with feature alignment is proposed to circumvent catastrophic forgetting. During inference, we introduce an entropy-based aggregation strategy to dynamically utilize the complementarity in the slow and fast learners. Extensive experiments on seven benchmark datasets verify the effectiveness of our method by significantly surpassing the state-of-the-art. Code will be available at https://github.com/MIFA-Lab/SAFE.

## 1   Introduction

Continual Learning (CL) requires deep learning models to incrementally incorporate new concepts from open-world data streams, while retaining previously learned knowledge. This presents a more challenging yet practical setting compared to traditional deep learning, which typically recognizes only closed-set categories. A variety of methods have been proposed for continual learning, including regularization-based [17, 20, 47], rehearsal-based [5, 14, 31], and dynamic network-based approaches [1, 43, 44]. These methods often assume that the model is trained from scratch, resulting in a substantial performance gap when compared to the joint training upper-bound.

---

✉Corresponding authors. *Equal contribution.

38th Conference on Neural Information Processing Systems (NeurIPS 2024).

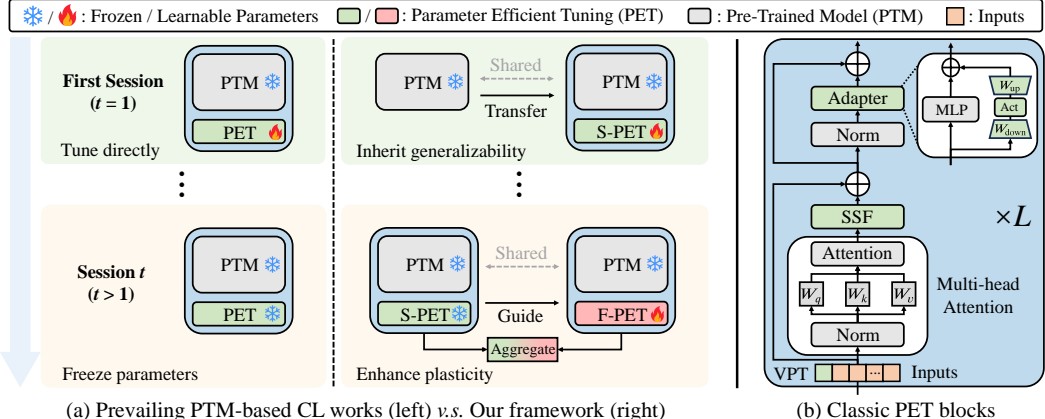

Figure 1: Comparisons of (a) prevailing PTM-based CL methods [2, 23, 52] and our Slow And Fast parameter-Efficient tuning (SAFE). The right part (b) illustrates several parameter-efficient tuning (PET) blocks: Adapter [6], Scale & Shift (SSF) [21], and Visual Prompt Tuning (VPT) [16].

Most recently, with the emergence of powerful pre-trained models, there has been growing interest in utilizing these foundational models as starting points for continual learning [23, 49, 52]. Pre-Trained Models (PTMs) which are often trained on vast datasets, encapsulate a wealth of general knowledge, effectively enhancing the performance of deep learning models in continual learning scenarios. As shown in the left part of Fig. 1(a), for adapting PTMs from pre-training datasets to continual learning datasets, prevailing works resort to parameter-efficient tuning (PET) techniques [6, 16, 21] in the first session. To restrain catastrophic forgetting, in incremental sessions, these works set parameters of the adapted model frozen [2, 15, 23, 52] and only update the classification weights in a training-free manner (*i.e.*, without gradient updates) to accommodate novel classes.

However, the above methods have two main limitations. First, direct parameter-efficient tuning in the first session will largely lose the general knowledge inherent in PTMs. This is because PTMs are pre-trained on a multitude of datasets while the dataset in the first session only contains relatively limited samples. Without proper transfer mechanisms, the knowledge from PTMs may be overwritten by the adapted model, which impedes the model's generalizability to unseen classes. Second, freezing parameters in the following sessions will hinder the plasticity of the model to further absorb new concepts not learned in the first session, resulting in a sub-optimal solution. Although several efforts have been made to mitigate the second limitation, existing works still face certain constraints such as additional storage requirement [35, 49], inferior online branch performance [8] and linearly increased model complexity [54].

Based on the above observations, in this paper, we propose Slow And Fast parameter-Efficient tuning (SAFE) to address existing challenges. In particular, SAFE demonstrates a unified framework that effectively inherits the generalizability of PTMs using slow parameter-efficient tuning (S-PET) and provides sufficient plasticity to learn task-specific knowledge in each incremental session using the fast one (F-PET). Meanwhile, SAFE does not require storing class distributions for data replay and only incurs constant-level additional computation and memory costs.

To achieve the above goals, SAFE employs distinct strategies for the first and subsequent sessions. In the first session, we focus on explicitly transferring general knowledge from pre-trained models (PTMs) by introducing a knowledge transfer loss. This involves computing a correlation matrix between feature embeddings from the PTM and the model with parameter-efficient tuning (PET). The diagonal elements of this matrix are maximized to ensure that the features remain consistent across both models, effectively aligning the PET-applied model's performance with that of the PTM. Simultaneously, minimizing the off-diagonal elements reduces redundancy in the embeddings, enhancing feature discriminability. After this tuning process, parameters can retain generalizable knowledge from the PTM. To prevent forgetting this knowledge, these trained parameters are subsequently frozen, with only the classification weights being updated, thus designating this model as the *slow* learner.

In the incremental sessions, to address the plasticity limitations of the slow learner, we introduce a *fast* learner capable of continuously integrating new concepts. Given the persistent challenge of catastrophic forgetting in continual learning, the slow learner guides the training of the fast learner. Concretely, we employ a feature alignment loss to minimize the distance between the embeddings of both learners on a hypersphere. Additionally, a cross-classification loss is proposed to ensure compatibility between the features of the fast learner and the classification weights of the slow learner, and vice versa. This approach allows the fast learner to assimilate new knowledge without storing exemplars or distributions, while also mitigating forgetting. For robust predictions, an entropy-based aggregation strategy is implemented during inference to dynamically leverage the complementary strengths of the slow and fast learners.

To summarize, the contributions of our paper are three-fold:

- To inherit the generalizable knowledge in PTMs that has been overlooked in existing continual learning works, we propose to explicitly transfer knowledge from the PTM to a slow learner. Once trained, the slow learner can generalize well to classes in incremental sessions.
- For improving the plasticity of CL models, we include a fast learner with guidance from the slow learner to continuously incorporate novel concepts. Moreover, by aggregating both slow and fast learners into a unified framework SAFE, robust predictions can be further made.
- The superiority of SAFE is validated on seven continual learning datasets where our method consistently achieves remarkable state-of-the-art performance. For example, our method surpasses the second-best result on ImageNet-A over $4\%$.

## 2 Related Work

**Continual Learning.** Traditional continual learning (CL) aims at continuously updating models with data streams from scratch. Existing strategies involve regularization-based approaches [17, 20, 47, 51] which prevent forgetting by regularizing network weights or predictions, rehearsal-based approaches which replay historical data stored in a fixed-sized buffer [4, 9, 14, 25, 31], and architecture-based approaches [1, 43, 44, 55] which dynamically expand models for novel classes. Among these methods, a recent attempt to preserve knowledge based on slow and fast complementary theory has been proposed [3, 22, 45]. Nevertheless, these approaches typically require adjusting all model parameters, which increases the computational burden of the learning process. Contrarily, our Slow And Fast parameter-Efficient tuning (SAFE) framework only requires much fewer learnable parameters as well as fewer resources, while obtaining more favorable performance.

**Continual Learning with Pre-Trained Models.** With the emergence of powerful pre-trained models (PTMs), it has become a hot topic to integrate pre-trained models with CL [29, 53] for better performance. Prompt-based methods [30, 33, 39, 41, 42] utilize prompt tuning to adapt PTMs to new tasks. However, these methods are tailored for Transformers [7, 37] and require an expanding prompt pool with the arrival of new data. First session adaptation methods [2, 23, 52] adapt PTMs solely in the first session and then freeze the model afterward to suppress forgetting [26, 32]. Nevertheless, these works lack plasticity for classes in subsequent sessions. Contrarily, another line of works focuses on continual adjustment [8, 35, 49, 54] to accommodate evolving information. However, the above approaches either require storing data distributions [35, 49] for replay, only obtain inferior online branch performance [8], or linearly increase complexity with incremental sessions [54]. Compared to existing works, our method provides a flexible framework that boosts generalizability by inheriting PTM's knowledge in the first session and maintains plasticity for incremental classes with constant complexity in a replay-free manner.

## 3 Method

### 3.1 Problem Definition

Following previous works [8, 23, 49, 52], in this paper, we mainly consider PTM-based CL under a class-incremental learning setting. Formally, the model is trained sequentially on a series of incremental sessions, where $\mathcal{D}^t = \{(x_i^t, y_i^t)\}_{i=1}^{N_t} \subset \{\mathcal{X}^t, \mathcal{Y}^t\}$ represents the $t$-th training set composed of $N_t$ samples, for $t \in \{1, 2, \dots, T\}$. The sample and label space of $\mathcal{D}^t$ are denoted by $\mathcal{X}^t$ and $\mathcal{Y}^t$, where $\mathcal{Y}^t$ is disjoint between different sessions, *i.e.*, $\forall i, j$ and $i \neq j$, $\mathcal{Y}^i \cap \mathcal{Y}^j = \varnothing$. We follow the

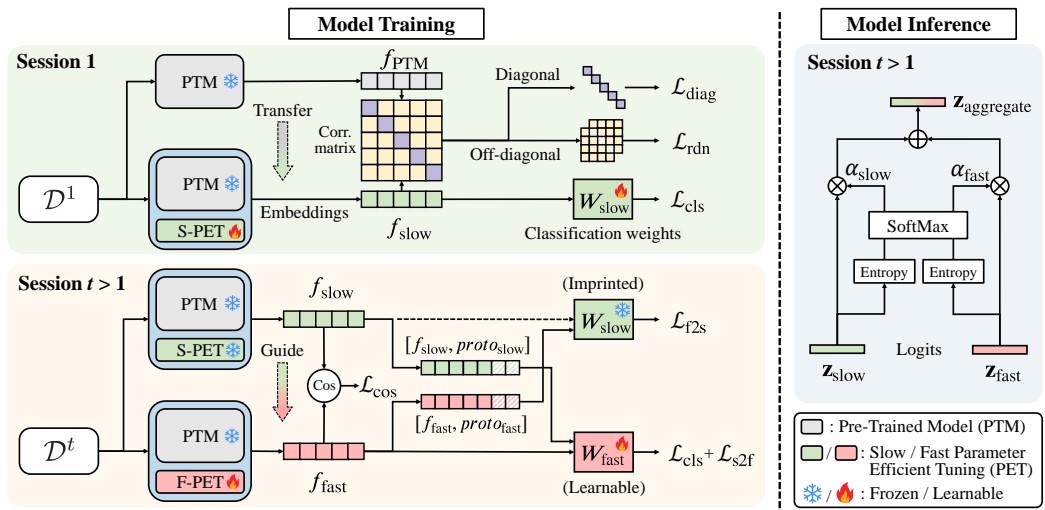

Figure 2: An overview of our SAFE framework. In the first session, PTM transfers knowledge to the slow learner for better generalization. In sessions $t > 1$, the fast learner is guided by the slow learner for enhanced plasticity. During inference, robust predictions are made by dynamic aggregation.

replay-free setting, where only $\mathcal{D}^t$ is accessible in session $t$. After training in the $t$-th session, the model is evaluated on all the seen classes so far: $\mathcal{Y}^{1:t} = \mathcal{Y}^1 \cup \mathcal{Y}^2 \cdots \cup \mathcal{Y}^t$. In addition, we also validate our method on domain-incremental learning setting, where the data distribution between sessions shifts significantly, *i.e.*, $\forall i, j$ and $i \neq j$, $P(\mathcal{X}^i) \neq P(\mathcal{X}^j)$, $\mathcal{Y}^i = \mathcal{Y}^j$.

## 3.2 Overall Architecture

For tackling the stability-plasticity dilemma in CL, we draw inspiration from the *complementary learning systems* theory [19] to develop a Slow And Fast parameter-Efficient tuning (SAFE) framework, as depicted in Fig. 2. In the first session, the slow learner is tuned to inherit the general knowledge from PTM and is frozen afterward. In the following sessions, the slow learner only updates its classification head using imprinted weights [28], which acts like the *neocortex* to slowly incorporate novel knowledge without forgetting. Complementary to this, the fast learner with learnable parameters rapidly encodes novel information as the *hippocampus* for adapting to new classes.

Formally, features extracted from PTM, slow learner and fast learner are denoted as $f_l = \phi_l(x) \in \mathbb{R}^d$, where $l \in \{\text{PTM}, \text{slow}, \text{fast}\}$ and $d$ is the feature dimension. To leverage the knowledge of PTMs with few learnable parameters and resources, feature extractors for the slow and fast learners are trained using parameter-efficient tuning (PET) [6, 16, 21] which are referred to as S-PET and F-PET, respectively. Consistent with prior works [23, 52, 54], we mainly consider three types of PETs: Adapter [6], SSF [21], and VPT [16], shown in the right part of Fig. 1.

The classification weights in session $t$ for the slow and fast learners are symbolized by $W_l \in \mathbb{R}^{d \times |\mathcal{Y}_{1:t}|}$, $l \in \{\text{slow}, \text{fast}\}$, where $|\mathcal{Y}_{1:t}|$ is the number of classes seen so far from session 1 to session $t$. For the slow learner, $W_{\text{slow}}$ is learned in the first session and expanded using feature centroids of training samples within the same classes [28] afterward to preserve learned general knowledge. Contrarily, $W_{\text{fast}}$ is trainable as CL progresses for the plasticity purpose.

In the following sections, we provide the details of slow and fast learner training in Section 3.3 and Section 3.4. After that, discussions about model inference are presented in Section 3.5.

## 3.3 Slow Learner

Benefiting from pre-training on large-scale resources, pre-trained models (PTMs) inherently possess strong generalizability for downstream tasks. Previous works [23, 35, 52] typically view the PTM as a preferable starting point for continual learning. To bridge the distribution gap between pre-training datasets and downstream datasets, these methods often directly apply PET to PTMs.

However, without proper transfer mechanisms, models directly tuned on downstream data cannot effectively inherit the general knowledge from PTMs. More seriously, the intrinsic knowledge in PTM may be overwritten during adaptation to the recent dataset, since it often contains relatively limited samples. To solve the above issues, we propose to effectively squeeze out information from PTMs and explicitly transfer it to adapted models.

Concretely, in the first session, we calculate the cross-correlation matrix $M \in \mathbb{R}^{d \times d}$ between the features of the slow learner and the PTM:

$$M_{i,j} = \frac{1}{N_b} \sum_{k=1}^{N_b} [\phi_{\text{PTM}}(x_k)]_i \cdot [\phi_{\text{slow}}(x_k)]_j, \tag{1}$$

where $N_b$ is the batch size, $d$ is the feature dimension and "$\cdot$" denotes element-wise multiplication[1]. Moreover, $i$ and $j$ index the dimensions of the features and matrices. In fact, the correlation matrix characterizes the relationship between feature embeddings of PTM and the slow learner. The $i$-th row and $j$-th column of $M$ measures the correlation between the $i$-th feature dimension (also termed as channel or pattern in the literature) of the PTM and the $j$-th feature dimension of the slow learner.

To encourage the PET-applied model to mimic the performance of the PTM, we maximize the elements in the diagonal. This maximizing term ensures the slow learner can learn invariant feature components that match the statistics of the PTM:

$$\mathcal{L}_{\text{diag}} = \frac{1}{d} \sum_{i=1}^{d} (1 - M_{i,i})^2. \tag{2}$$

Additionally, we reduce the redundancy between patterns in embeddings to enhance discriminability. This can obtained by decreasing the off-diagonal elements in $M$ with $\mathcal{L}_{\text{rdn}}$:

$$\mathcal{L}_{\text{rdn}} = \frac{1}{d \cdot (d-1)} \sum_{i=1}^{d} \sum_{j \neq i} M_{i,j}^2. \tag{3}$$

Combined with the classification loss $\mathcal{L}_{\text{cls}}$ using cross-entropy (CE):

$$\mathcal{L}_{\text{cls}} = \frac{1}{N_b} \sum_{i=1}^{N_b} \text{CE}(W_{\text{slow}}^{\top} \odot \phi_{\text{slow}}(x_i), y_i), \tag{4}$$

where "$\odot$" denotes matrix multiplication, the overall loss function during the first training session is defined as:

$$\mathcal{L}_{\text{slow}} = \mathcal{L}_{\text{cls}} + \lambda_{\text{diag}} \cdot \mathcal{L}_{\text{diag}} + \lambda_{\text{rdn}} \cdot \mathcal{L}_{\text{rdn}}. \tag{5}$$

In Eq. (5), $\lambda_{\text{diag}}$ and $\lambda_{\text{rdn}}$ are the balancing hyper-parameters. Intuitively, the joint optimization of three losses makes the adapted model simultaneously acquire distribution-specific knowledge based on $\mathcal{L}_{\text{cls}}$ and inherit general knowledge of the PTM using $\mathcal{L}_{\text{diag}}$ and $\mathcal{L}_{\text{rdn}}$. As a result, the slow model can better generalize to incoming classes even unseen in the first training session.

### 3.4 Fast Learner

Although solely using the slow learner with general features already obtains competitive performance, the plasticity of the model is hindered due to its frozen parameters in the following sessions. To strike a balance between stability and plasticity, we adopt the fast learner to continuously learn episodic information for novel classes. However, updating representations without data reply will lead to semantic drift [35, 46, 49], causing catastrophic forgetting of previously learned knowledge. Existing works to address this problem either store additional data distributions [35, 49] or require sophisticated drift estimations after each session [35, 46]. Compared to previous works, our method imposes no such constraints, and aligns the models before and after updates in a single embedding space, essentially addressing semantic drift.

---

[1]Following [14], in this paper, we use $l_2$ normalization to map features and classification weights onto a hypersphere before element-wise or matrix multiplication. Normalization is omitted to simplify notation.

First, the fast learner is trained with guidance from the slow learner using feature alignment to preserve prior representations. Specifically, the distance of feature embedding from both models is minimized on a hypersphere to alleviate forgetting:

$$\mathcal{L}_{\text{cos}} = \frac{1}{N_b} \sum_{i=1}^{N_b} \left(1 - \cos(\phi_{\text{slow}}(x_i), \phi_{\text{fast}}(x_i))\right), \tag{6}$$

where $N_b$ is the batch size and $\cos$ denotes cosine similarity of two vectors.

Furthermore, we utilize cross-classification which contains a fast-to-slow loss and a slow-to-fast loss to maintain previous decision boundaries. For fast-to-slow calibration, we feed features from the fast learner to the classification layer of the slow learner. This objective makes features from the fast model compatible with the decision boundaries of the slow one to suppress semantic drift. Moreover, since the classification weight vector of each class can be viewed as a *prototype* of that class [28, 34], we also use these vectors as inputs for further preserving knowledge from previous sessions:

$$\mathcal{L}_{\text{f2s}} = \frac{1}{N_b} \sum_{i=1}^{N_b} \text{CE}(W_{\text{slow}}^\top \odot \phi_{\text{fast}}(x_i), y_i) + \frac{1}{|\mathcal{Y}_{1:t-1}|} \sum_{j=1}^{|\mathcal{Y}_{1:t-1}|} \text{CE}(W_{\text{slow}}^\top \odot W_{\text{fast}}^{(j)}, j), \tag{7}$$

where $W_{\text{fast}}^{(j)} \in \mathbb{R}^d$ denotes the $j$-th column of $W_{\text{fast}}$, which is also the prototype for class $j$ in the fast learner. Similarly, slow-to-fast loss $\mathcal{L}_{\text{s2f}}$ can be derived by swapping the fast and slow terms in Eq. (7). After that, the cross-classification loss can be defined as $\mathcal{L}_{\text{s}\leftrightarrow\text{f}} = \mathcal{L}_{\text{f2s}} + \mathcal{L}_{\text{s2f}}$.

Along with the classification loss $\mathcal{L}_{\text{cls}}$ in Eq. (4) applied to the fast learner, the optimization objective in the incremental phase is defined as:

$$\mathcal{L}_{\text{fast}} = \mathcal{L}_{\text{cls}} + \mathcal{L}_{\text{s}\leftrightarrow\text{f}} + \lambda_{\text{cos}} \cdot \mathcal{L}_{\text{cos}}, \tag{8}$$

where $\lambda_{\text{cos}}$ is the balancing hyper-parameter. The loss function $\mathcal{L}_{\text{fast}}$ smoothly adapts the fast learner to new knowledge while enforcing consistency with previously acquired knowledge, which boosts the plasticity of the model without severe forgetting.

## 3.5 Model Inference

Since the slow learner inherits general knowledge and the fast learner contains task-adaptive knowledge, we can obtain robust predictions by utilizing the complementarity of them. We first introduce the inference using a single learner and then provide aggregation strategy based on both learners.

**Single-learner-based Inference.** Following previous work [23, 26], instead of directly using the classification weights $W_l$ and features $\phi_l(x)$, $l \in \{\text{slow}, \text{fast}\}$ for prediction, we take advantage of second-order statistics and prototype information for better performance. Formally, given a test sample $x$, the predicted logits of each learner $\mathbf{z}_l$ are calculated as:

$$\mathbf{z}_l = \tilde{W}_l^\top \odot (G + \beta I)^{-1} \odot h_l(x) \in \mathbb{R}^{|\mathcal{Y}_{1:t}|}, \tag{9}$$

where $\beta$ is a hyper-parameter for regularization, $h_l(x) \in \mathbb{R}^M$ is projected feature of $x$ and classification weights $\tilde{W}_l \in \mathbb{R}^{M \times |\mathcal{Y}_{1:t}|}$ is composed of summations of projected features with same class labels. Gram matrix $G \in \mathbb{R}^{M \times M}$ is cumulated based on training data from session 1 to $t$:

$$G = \sum_{s=1}^{t} \sum_{i=1}^{N_t} h_l(x_i^s) \odot h_l(x_i^s)^\top, \ h_l(x_i^s) = \psi(W_{\text{rand}}^\top \odot \phi_l(x_i^s)). \tag{10}$$

In Eq. (10), $W_{\text{rand}} \in \mathbb{R}^{d \times M}$ is the projection matrix with each column sampled from $\mathcal{N}(0, \sigma^2 I)$, $\psi$ is a nonlinear activation function and $I$ denotes the identity matrix. Mathematically, Eq. (9) defines a more general form of regular linear prediction. When $W_{\text{rand}}$ is $I$ and $\psi$ is not applied, it degrades to a ridge regression [13]. Moreover, if $G$ is removed, the classifier further reduces to NCM [24].

**Aggregation-based Inference.** As discussed in the above sessions, slow and fast learners excel in handling classes from different sessions. Due to its plasticity, the fast learner can better recognize categories from the latest several sessions but shows limited performance on the old ones caused by potential forgetting. Contrarily, despite limited novel concept adaptation, the slow learner can capture historical knowledge thanks to its stability. Intuitively, when dealing with proficient categories, the

model exhibits higher confidence in predictions. Motivated by this, we use entropy to measure the confidence and dynamically aggregate the logits for robust predictions.

Given a test sample, we compute the entropy of predictions using $\mathcal{H} = -\sum_i p_i \log p_i$ for each learner, obtaining $\mathcal{H}_{\text{slow}}$ and $\mathcal{H}_{\text{fast}}$, where $p = \text{softmax}(\mathbf{z})$ is predicted probability. As lower entropy indicates less uncertainty in predictions, the confidence of each learner can be represented by $[\alpha_{\text{slow}}, \alpha_{\text{fast}}] = \text{softmax}([-\gamma \cdot \mathcal{H}_{\text{slow}}, -\gamma \cdot \mathcal{H}_{\text{fast}}])$, where $\gamma$ is a scalar to control the peakiness of output distributions. After that, the aggregated logits $\mathbf{z}_{\text{aggregate}}$ automatically assign higher weights to predictions with higher confidence, and can be obtained using a convex combination:

$$\mathbf{z}_{\text{aggregate}} = \alpha_{\text{slow}} \cdot \mathbf{z}_{\text{slow}} + \alpha_{\text{fast}} \cdot \mathbf{z}_{\text{fast}}. \tag{11}$$

Finally, the prediction is obtained using the index of the max element in $\mathbf{z}_{\text{aggregate}}$ in session $t > 1$, while using $\mathbf{z}_{\text{slow}}$ instead in session 1 since the fast learner is not available in that session.

## 4 Experiments

In this section, we first introduce the implementation details of our proposed method SAFE and then compare it to the state-of-the-art on seven popular benchmark datasets. After that, detailed ablative experiments are conducted to validate the effectiveness of each component.

### 4.1 Experimental Setups

**Datasets and Evaluation.** Following previous methods [23, 52, 54], our evaluations are conducted on seven benchmark datasets: CIFAR100 [18], ImageNet-R (IN-R) [11], ImageNet-A (IN-A) [12], CUB200 [38], Omnibenchmark (OB) [50], VTAB [48] and DomainNet [27]. Previous state-of-the-art PTM-based CL methods are chosen for comparison, including L2P [42], DualPrompt [41], CODAPrompt [33], ADaM [52], RanPAC [23], SSIAT [35], and SLCA [49]. We adopt final accuracy $\text{Acc}_T$ and average accuracy $\text{Acc}_{avg} = \frac{1}{T} \sum_{t=1}^{T} \text{Acc}_t$ as evaluation metrics.

**Implementation Details.** Consistent to existing works [23], we adopt ViT-B/16-IN1K and ViT-B/16-IN21K as the PTM and apply Adapter [6], SSF [21] or VPT [16] for parameter-efficient tuning (Appendix A). In each session, we train the model for 20 epochs using SGD optimizer, weight decay of 0.0005, momentum of 0.9, and a cosine annealing schedule where learning rate starts from 0.01 and decays to 0. The batch size is set to 48. In addition, $\beta$ in Eq. (9) is selected based on the performance on the training data similar to [23]. For other hyper-parameters used in our method, we find $\lambda_{\text{diag}} = 0.1$, $\lambda_{\text{rdn}} = 100$, $\lambda_{\cos} = 50$, $\gamma = 1$ is a reasonable set of default choices. Detailed hyper-parameter sensitivity analyses are provided in Appendix D.

### 4.2 Comparisons with State-of-The-Arts

In this section, we compare the proposed method SAFE with several state-of-the-art approaches across seven datasets: CIFAR100, ImageNet-R, ImageNet-A, Omnibenchmark, CUB200, VTAB and DomainNet. For fairness, all methods are implemented with the same ViT [7] backbones.

The class-incremental learning results from the final session are reported in Table 2. As shown in Table 2, our method consistently achieves the best performance among all benchmarks. Notably, we significantly surpass the second-best result on ImageNet-A by $4.4\%$. When compared to methods storing additional data distributions for replay [35, 49], our method is replay-free and can still outperform these methods by a significant margin. In addition, we improve the average accuracy over six datasets by $2.1\%$ compared to the previous best approach [23]. The aforementioned superiority can contribute to the generalizability and plasticity of our method within a unified framework.

Table 1: Performance on DomainNet.

| Method | Final Acc. |
|---|---|
| L2P [42] | 40.2 |
| S-iPrompts [40] | 50.6 |
| ADaM [52] | 50.3 |
| RanPAC [23] | 66.6 |
| Slow learner | 67.04 |
| Fast learner | 67.49 |
| SAFE (ours) | **67.82** |

For domain-incremental learning, results on DomainNet with 6 different domains are summarized in Table 1. Our method SAFE can outperform the second-best result by $1.2\%$, demonstrating that the

proposed framework is applicable to scenarios where the data distribution of the first task diverges significantly from that of subsequent tasks.

Table 2: Performance comparisons on six class-incremental learning datasets. The final accuracy (%) of each dataset is reported in the table, and the last column presents the averaged accuracy over all the datasets. Methods with/without data replay are noted using "w/" and "w/o", respectively.

| Method | Replay | CIFAR | IN-R | IN-A | CUB | OB | VTAB | Avg |
|---|---|---|---|---|---|---|---|---|
| SLCA [49] | w/ | 91.5 | 77.0 | 59.8 | 84.7 | 73.1 | 89.2 | 79.2 |
| SSIAT [35] | | 91.4 | 79.6 | 62.2 | 88.8 | - | 94.5 | - |
| L2P [42] | | 84.6 | 72.5 | 42.5 | 65.2 | 64.7 | 77.1 | 67.8 |
| DualPrompt [41] | | 81.3 | 71.0 | 45.4 | 68.5 | 65.5 | 81.2 | 68.8 |
| CODAPrompt [33] | w/o | 86.3 | 75.5 | 44.5 | 79.5 | 68.7 | 87.4 | 73.7 |
| ADaM [52] | | 87.6 | 72.3 | 52.6 | 87.1 | 74.3 | 84.3 | 76.4 |
| EASE [54] | | 87.8 | 76.2 | 55.0 | 86.8 | 74.9 | 93.6 | 79.1 |
| RanPAC [23] | | 92.2 | 78.1 | 61.8 | 90.3 | 79.9 | 92.6 | 82.5 |
| SAFE (ours) | w/o | **92.8** | **81.0** | **66.6** | **91.1** | **80.9** | **95.0** | **84.6** |

## 4.3 Ablation Study

To investigate the factors contributing to the success of SAFE, we validate the effectiveness of our key components: the slow learner (SL) in Section 3.3, the fast learner (FL) in Section 3.4, and the entropy-based aggregation in Section 3.5. Experiments are primarily conducted on IN-A dataset.

**Effectiveness of the Slow Learner.** We assess the effectiveness of the slow learner from three perspectives. Firstly, as depicted in Table 3, when the slow learner is added to the baseline [23], the final accuracy increases by 3.2% and the average accuracy increases by 2.1%. This observation verifies that the slow learner can generalize well to the incremental classes.

Secondly, we expect the slow learner to inherit generalizability from the PTM. To dive deeper into this aspect, we visualize the embeddings of five unseen classes and five seen classes by T-SNE [36] after the first session adaptation. As shown in Fig. 3, the embedding space of the slow learner exhibits distinct separation between the seen and unseen classes. Note that the feature distributions with SL in the grey ellipse become more separable compared with the baseline method. This illustrates the successful integration of generalization capabilities from the PTM into the slow learner.

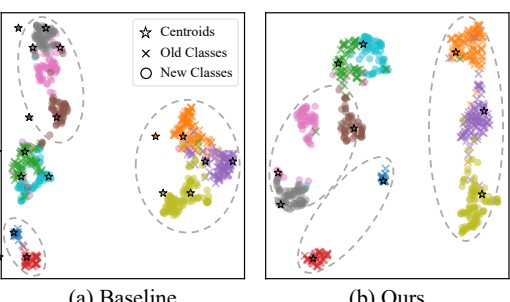

(a) Baseline    (b) Ours

Figure 3: Comparisons with T-SNE visualization.

Furthermore, we explore other alternatives for transferring generalizability, including feature alignment (FA) by distilling PTM's features, logits alignment (LA) by distilling PTM's predictions, and second-order statistics alignment (SSA) by distilling PTM's covariance. Table 5 presents the average and final accuracy of the substitutions on IN-A, with the best results highlighted in bold. It is observed that our slow learner can consistently outperform these variations, validating its superiority.

**Effectiveness of the Fast Learner.** As shown in the third row of Table 3, compared to the slow learner, using only the fast learner can obtain 1.1% improvements in the final accuracy. This indicates

Table 3: Overall ablation study on IN-A.

| Method | Final | Avg |
|---|---|---|
| Baseline | 62.21 | 72.31 |
| Slow Learner | 65.44 | 74.41 |
| Fast Learner | 66.49 | 74.50 |
| Slow & Fast Learner (SAFE) | **66.56** | **74.71** |

Table 4: Ablation study of aggregation.

| Method | Final | Avg |
|---|---|---|
| Features Concatenate | 65.59 | 73.22 |
| Logits Add | 65.90 | 73.31 |
| Logits Max | 66.03 | 73.46 |
| Entropy-based Aggregate | **66.56** | **74.71** |

Table 5: Ablation study of the slow learner.

| Method | Final | Avg |
|---|---|---|
| Baseline | 62.21 | 72.31 |
| Baseline + FA | 62.81 | 73.35 |
| Baseline + LA | 64.06 | 73.70 |
| Baseline + SSA | 63.20 | 73.00 |
| Baseline + $\mathcal{L}_{\text{slow}}$ (Slow Learner) | **65.44** | **74.41** |

Table 6: Ablation study of the fast learner.

| Method | FT | $\mathcal{L}_{\text{s}\leftrightarrow\text{f}}$ | $\mathcal{L}_{\text{cos}}$ | Final | Avg |
|---|---|---|---|---|---|
| Baseline | | | | 62.21 | 72.31 |
| Finetune directly | ✓ | | | 8.16 | 30.73 |
| Finetune w/ $\mathcal{L}_{\text{s}\leftrightarrow\text{f}}$ | ✓ | ✓ | | 65.31 | 73.88 |
| Finetune w/ $\mathcal{L}_{\text{cos}}$ | ✓ | | ✓ | 66.07 | 74.20 |
| Fast Learner | ✓ | ✓ | ✓ | **66.49** | **74.50** |

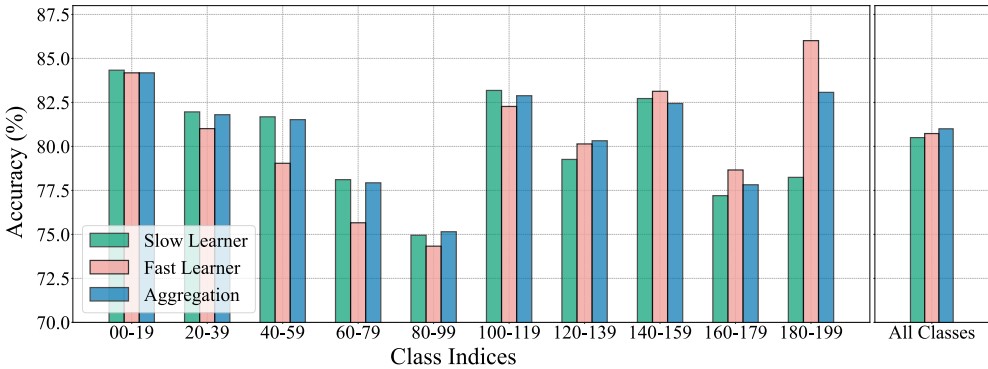

Figure 4: Validations on the necessity of the aggregation on IN-R. We provide detailed classification accuracy of test samples from different sessions. Results of the slow learner, the fast learner and SAFE are presented for comparison.

that the fast learner is properly guided by the slow learner, and thus can continuously adapt to novel classes with suppressed forgetting.

Subsequently, we present the necessity of each regularization term in the fast learner. As shown in Table 6, without $\mathcal{L}_{\text{s}\leftrightarrow\text{f}}$ and $\mathcal{L}_{\text{cos}}$, the performance drops to lower than 10% due to catastrophic forgetting. To alleviate forgetting, both $\mathcal{L}_{\text{s}\leftrightarrow\text{f}}$ and $\mathcal{L}_{\text{cos}}$ are applied. Specifically, solely using $\mathcal{L}_{\text{s}\leftrightarrow\text{f}}$ results in an improvement of 3.1% compared to the baseline, while using only $\mathcal{L}_{\text{cos}}$ yields a gain of 3.9% over the baseline. Moreover, with all the proposed loss functions, the fast learner can obtain the best performance, validating the effectiveness of each regularization term.

**Effectiveness of Aggregation.** As shown in the last row of Table 3, the combination of the slow and fast learners presents the best result. This observation is consistent with the complementary learning systems theory [19] that memory necessitates the presence of both a slow learner and a fast learner for improved performance.

To gain deeper insights into the necessity of both learners, we elaborate on their final accuracy of classes from each session. In Fig. 4, the slow learner, mimicking the neocortex, initially stores structured information and performs well on relatively old classes (0-119). Conversely, the fast learner, resembling the hippocampus, swiftly adapts novel concepts and excels in more recent classes (120-199). From this perspective, combining these two complementary learners leverages their strengths across the training process, resulting in superior model performance.

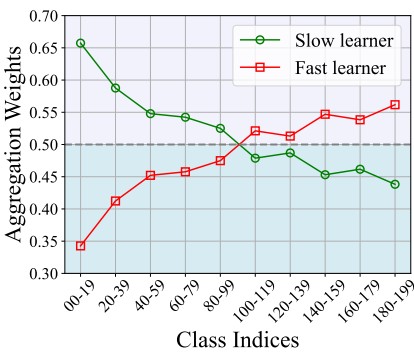

Figure 5: Aggregation weights for the slow learner and fast learner on IN-R.

In addition, Fig. 5 illustrates how the aggregated model dynamically leverages the strengths of both learners. Concretely, the horizontal axis represents the class indices to which each test sample belongs, while the vertical axis shows the average aggregation weights of each learner assigned to these test samples. It is observed from Fig. 5 that, for classes 120-199, the fast learner consistently shows higher weights, which is consistent with its superior classification accuracy in these classes as

depicted in Fig. 4. For classes 0-119, the slow learner obtains higher weights, generally aligning with its demonstrated stability and better performance on these classes shown in Fig. 4. By adaptively balancing the contributions of both learners, our method achieves a harmonious trade-off between stability and adaptability.

Moreover, we undertake detailed comparisons to other merging strategies to validate the effectiveness of our aggregation choice. As depicted in Table 4, we compare our entropy-based aggregation with three alternatives: feature concatenation, logits addition, and logits max. We report the final and average accuracy, where the results elucidate that the entropy-based aggregation fully leverages both learners and achieves the best performance.

### 4.4 Memory Usage

In this section, we investigate the number of learnable parameters in different methods and report the parameter-performance comparison. Since no exemplars are stored in our method, the primary storage cost is attributed to the trainable model parameters introduced by parameter-efficient tuning (PET). Although PET entails additional parameters, it is still small relative to the overall size of the pre-trained model (PTM). Moreover, as the parameter-performance trade-off shown in Fig. 6, our method SAFE utilizes a similar scale of parameters as existing PTM-based methods while achieving substantial performance improvements.

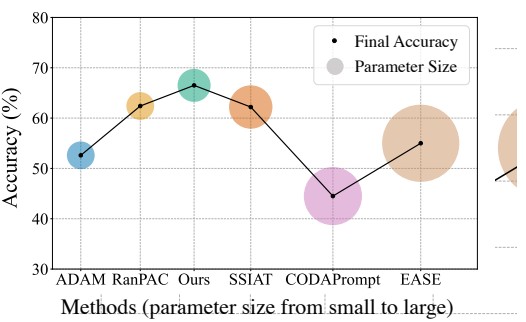

Figure 6: Memory usage comparison.

## 5 Conclusion

In this paper, we introduced SAFE, a Slow And Fast parameter-Efficient tuning framework for continual learning. Our approach leverages the inherent knowledge in pre-trained models (PTMs) while maintaining model plasticity for novel concepts. By incorporating a transfer loss function, we ensure the preservation of general knowledge from PTMs. In the first session, we calibrate slow efficient tuning parameters to enhance the model's ability to generalize to new classes. To balance stability and plasticity, we fix the slow efficient tuning parameters and continuously update the fast ones, employing a cross-classification loss with feature alignment to prevent catastrophic forgetting. During inference, we introduce an entropy-based aggregation strategy for dynamic utilization of the complementarity between the slow learner and the fast learner. Extensive experiments on seven benchmark datasets demonstrate that our method significantly surpasses the state-of-the-art, validating the effectiveness of our approach.

**Limitations:** Our approach is built upon RanPAC [23], and as such, it shares some of the same limitations. For instance, our method relies on a strong feature extractor to effectively inherit generalizability from PTMs, making it less suitable for scenarios where training needs to be performed from scratch or starting from rather small tasks. Additionally, our method introduces three hyper-parameters to balance the loss functions during training, as previously discussed. While our experiments demonstrate that a set of default values works well across the benchmark datasets evaluated in our work, we acknowledge that these choices might not be optimal when applied to datasets with essentially different statistical characteristics. Furthermore, slowly updating the slow learner periodically, rather than keeping it fixed in subsequent sessions, may further enhance the model's adaptability and could be a promising direction for future research.

## Acknowledgement

This project was funded by National Natural Science Foundation of China (62406192). The project and raw idea were initiated at MIFA Lab of Shanghai Jiao Tong University, while the major part of the work was completed at Tencent YouTu Lab.

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

# Appendices

## A    Parameter-Efficient Tuning (PET)

Trained on vast amounts of data, models with billions of parameters exhibit remarkable performance across various tasks. However, the expansive scale and computation pose considerable challenges when customizing them for downstream deployment. From this perspective, parameter-efficient tuning (PET) provides a practical solution by effectively adapting the large pre-trained models with limited additional parameters.

Parameter-efficient tuning selectively adjusts a small proportion of the model parameters while keeping the rest frozen. In this way, pre-trained models (PTMs) can partially keep the generalization and deal with domain gaps at a low resource cost [10]. Motivated by this, PTM-based continual learning models leverage PET in their paradigm to achieve desirable results [23, 52, 54]. Typically, continual learning works leverage ViT-B/16-IN1K and ViT-B/16-IN21K as the backbones and fine-tune the model mainly with three PET algorithms: Adapter [6], Scale & Shift (SSF) [21] and Visual Prompt Tuning (VPT) [16], which are introduced in the following:

**Adapter:** Adapters are small additional layers inserted into the layers of a PTM. Each adapter layer generally consists of three parts: a down-projection layer $W_{\text{down}} \in \mathbb{R}^{d \times r}$ which reduces the input feature dimension, a non-linear activation function (*e.g.*, ReLU), and an up-projection layer $W_{\text{up}} \in \mathbb{R}^{r \times d}$ which projects features back to the original dimension. Specifically, given an input $\boldsymbol{x} \in \mathbb{R}^{L \times d}$, the output $\boldsymbol{y} \in \mathbb{R}^{L \times d}$ is expressed as:

$$\boldsymbol{y} = \text{MLP}(\boldsymbol{x}) + \text{ReLU}(\boldsymbol{x} \odot W_{\text{down}}) \odot W_{\text{up}}, \tag{12}$$

where $L$, $d$ and $r$ represent the length of the input feature sequence, original feature dimension and projected feature dimension. In the above equation, "$\odot$" denotes matrix multiplication.

**Scale & Shift (SSF):** SSF involves two main operations: scaling, which multiplies each feature by a learnable vector to adjust its spread, and shifting, which adds a trainable vector to each feature to change its central position. In the context of fine-tuning PTMs, SSF helps to normalize the feature distributions and adjust to new data. This improves performance and robustness by maintaining consistency in distribution. Specifically,

$$\boldsymbol{y} = \gamma \cdot \boldsymbol{x} + \beta, \tag{13}$$

where $\gamma \in \mathbb{R}^d$ and $\beta \in \mathbb{R}^d$ are the scaling and shifting vectors, respectively. Moreover, "$\cdot$" represents element-wise multiplication.

**Visual Prompt Tuning (VPT):** VPT extends original input features with lightweight learnable tokens and the extended features will be fed into subsequent transformer blocks of ViT [7] to obtain the final adapted embedding. Concretely, denote the learnable prompts as $\boldsymbol{P} \in \mathbb{R}^{K \times d}$, extended features can be expressed as:

$$\boldsymbol{y} = [\boldsymbol{P}, \boldsymbol{x}], \tag{14}$$

where $K$ is the length of the prompt and $\boldsymbol{y} \in \mathbb{R}^{(K+L) \times d}$ is the extended feature.

## B    Effects of PET to SAFE

In the main paper, we report the remarkable performance of the proposed SAFE framework under the same PET setting as [23]. In this section, we demonstrate that the proposed approach is a general framework that is compatible with diverse PET modules. Specifically, we combine SAFE with Adapter [6], Scale & Shift(SSF) [21] and Visual Prompt Tuning (VPT) [16]. As depicted in Table 7, we report the final accuracy on six datasets compared with the baseline method [23].

As shown in Table 7, the proposed SAFE framework outperforms the baseline across various PET modules by a substantial margin. It is worth noting that the proposed method consistently exceeds the baseline on ImageNet-A by over $4\%$ with different PET modules. We also achieve performance improvements by $2.2\%$ with Adapter, $3.1\%$ with SSF, and $3.0\%$ with VPT on ImageNet-R. These results demonstrate the general applicability of our framework across PET algorithms.

Table 7: Performances of SAFE and our baseline PanPAC [23] with three different parameter-efficient tuning (PET) modules on six datasets. The rows in shadow show improvements compared to the baseline. The best results are in **bold**.

| Method | PET | CIFAR | IN-R | IN-A | CUB | OB | VTAB | Avg |
|---|---|---|---|---|---|---|---|---|
| Baseline | | 92.2 | 77.8 | 59.9 | 90.3 | 79.6 | 92.6 | 82.1 |
| SAFE (ours) | Adapter | **92.8** | **80.0** | **64.1** | **91.1** | **80.3** | **94.3** | **83.8** |
| *Improve* | | +0.6 | +2.2 | +4.2 | +0.8 | +0.7 | +1.7 | +1.7 |
| Baseline | | 90.3 | 77.9 | 62.4 | 89.9 | 78.8 | 92.2 | 81.9 |
| SAFE (ours) | SSF | **91.6** | **81.0** | **66.6** | **91.0** | **79.8** | **95.0** | **84.2** |
| *Improve* | | +1.3 | +3.1 | +4.2 | +1.1 | +1.0 | +2.8 | +2.3 |
| Baseline | | 90.0 | 76.7 | 61.2 | 89.7 | 79.9 | 91.6 | 81.5 |
| SAFE (ours) | VPT | **92.2** | **79.7** | **65.7** | **90.8** | **80.9** | **93.4** | **83.8** |
| *Improve* | | +2.2 | +3.0 | +4.5 | +1.1 | +1.0 | +1.8 | +2.3 |

## C  Pseudo-code

For the detailed training procedure of the slow learner in Section 3.3 and the fast learner in Section 3.4, we summarize the pseudo-code of our method SAFE training in Algorithm 1.

---

**Algorithm 1** Model Training in Incremental Session $t$

---

**Input:** Model from session $t-1$, training data $\mathcal{D}^t$ from session $t$.
**Output:** Updated model in session $t$.
 1: **Phase 1:** Slow learner in session $t = 1$.
 2: Freeze pre-trained model parameters $\theta_{\mathrm{PTM}}$.
 3: Randomly initialize classification weights $W_{\mathrm{slow}}$ and efficient tuning parameters $\theta_{\mathrm{S\text{-}PET}}$.
 4: **while** not done **do**
 5:     $\{(x, y)\} \leftarrow$ sample a batch of data from $\mathcal{D}^1$.
 6:     Calculate the correlation matrix in Eq. (1) and losses $\mathcal{L}_{\mathrm{diag}}$, $\mathcal{L}_{\mathrm{rdn}}$ in Eq. (2), Eq. (3).
 7:     Calculate the overall loss function $\mathcal{L}_{\mathrm{slow}}$ in Eq. (5).
 8:     Update $\{W_{\mathrm{slow}}, \theta_{\mathrm{S\text{-}PET}}\}$ with gradients $\nabla \mathcal{L}_{\mathrm{slow}}$.
 9: **end while**
10: Replace $W_{\mathrm{slow}}$ with imprinted weights (*i.e.*, feature centroids of each class in $\mathcal{D}^1$).
11: Freeze parameters $\{W_{\mathrm{slow}}, \theta_{\mathrm{S\text{-}PET}}\}$.
12:
13: **Phase 2:** Fast Learner in session $t > 1$.
14: Expand $W_{\mathrm{slow}}(\mathbb{R}^{d \times |\mathcal{Y}_{1:t-1}|} \to \mathbb{R}^{d \times |\mathcal{Y}_{1:t}|})$ with imprinted weights using $\phi_{\mathrm{slow}}$ and $\mathcal{D}^t$.
15: Expand $W_{\mathrm{fast}}(\mathbb{R}^{d \times |\mathcal{Y}_{1:t-1}|} \to \mathbb{R}^{d \times |\mathcal{Y}_{1:t}|})$ with imprinted weights using $\phi_{\mathrm{fast}}$ and $\mathcal{D}^t$.
16: Initialize the fast learner's efficient tuning parameters $\theta_{\mathrm{F\text{-}PET}}$ from session $t-1$.
17: **while** not done **do**
18:     $\{(x, y)\} \leftarrow$ sample a batch of data from $\mathcal{D}^t$.
19:     Calculate feature alignment loss $\mathcal{L}_{\mathrm{cos}}$ in Eq. (6) and cross-classification loss $\mathcal{L}_{\mathrm{s \leftrightarrow f}}$ in Eq. (7).
20:     Calculate the overall loss function $\mathcal{L}_{\mathrm{fast}}$ in Eq. (8).
21:     Update $\{W_{\mathrm{fast}}, \theta_{\mathrm{F\text{-}PET}}\}$ with gradients $\nabla \mathcal{L}_{\mathrm{fast}}$.
22: **end while**

---

## D  Further Ablations

**Hyper-Parameters Sensitivity.** Our framework SAFE includes 4 hyper-parameters: $\lambda_{\mathrm{diag}}$ and $\lambda_{\mathrm{rdn}}$ for the slow learner, $\lambda_{\mathrm{cos}}$ for the fast learner, and $\gamma$ for aggregation. In this section, we supply detailed hyper-parameter sensitivity analyses on ImageNet-A. Results for $\lambda_{\mathrm{diag}}$ and $\lambda_{\mathrm{rdn}}$ are depicted in Fig. 7, while the results for $\lambda_{\mathrm{cos}}$ are shown in Table 8. Moreover, Table 9 presents the experiment on $\gamma$. It is observed that hyper-parameters remain relatively stable within a certain range. For example, the slow learner can achieve satisfactory results with $\lambda_{\mathrm{diag}}$ in the range from 0.1 to 1, and $\lambda_{\mathrm{rdn}}$ in the range from 100 to 500. The fast learner can obtain good performance with $\lambda_{\mathrm{cos}}$ in the interval from

50 to 100. Moreover, the aggregation module works well by simply setting $\gamma$ to 1. As a result, we set $\lambda_{\mathrm{diag}} = 0.1$, $\lambda_{\mathrm{rdn}} = 100$, $\lambda_{\mathrm{cos}} = 50$, $\gamma = 1$ as the default choices of hyper-parameters as stated in Section 4.1 of our main paper.

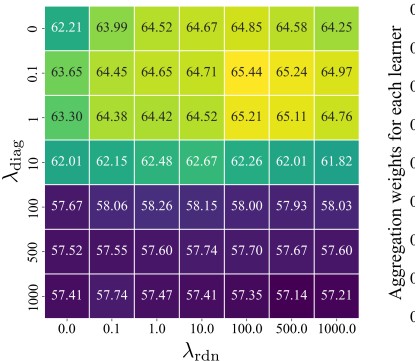

Figure 7: Ablations of hyper-parameter sensitivity on $\lambda_{\mathrm{diag}}$ and $\lambda_{\mathrm{rdn}}$ for the slow learner.

Table 8: Ablation of $\lambda_{\mathrm{cos}}$ on the fast learner.

| $\lambda_{\mathrm{cos}}$ | 0 | 0.1 | 1 | 10 | 50 | 100 |
|---|---|---|---|---|---|---|
| FL | 18.56 | 21.33 | 40.49 | 65.20 | **66.49** | 66.08 |

Table 9: Ablation of $\gamma$ on the aggregated model.

| $\gamma$ | 0 | 0.1 | 1 | 5 | 10 | 100 |
|---|---|---|---|---|---|---|
| SAFE | 65.90 | 66.36 | **66.56** | 66.50 | 66.24 | 66.03 |

**Teacher Models for the Fast Learner.** As discussed in Section 3.4, the fast learner is guided by the slow learner during adapting to novel classes. In this section, we provide additional experiments on the choice of the teacher model which guides the training of the fast learner. We conduct comparisons on training the fast learner directly (None teacher), using the pre-trained model as a teacher (PTM) and using the fast learner from the last session as a teacher ($t-1$). As shown in Table 10, utilizing the slow learner as a teacher model surpasses all the alternatives. This is because the slow learner can provide generalizable knowledge to the fast learner and simultaneously alleviate forgetting.

Table 10: Ablation of the teacher model for the fast learner.

| Teacher | Fast learner | |
|---|---|---|
| | Final | Avg |
| None | 8.16 | 30.73 |
| PTM | 55.76 | 65.46 |
| Fast learner ($t-1$) | 63.66 | 74.25 |
| Slow learner | **66.49** | **74.50** |

# E    Comparisons with RanPAC

While both the proposed SAFE and PanPAC [23] leverage PTMs for continual learning, they target different components of the model. Specifically, RanPAC focuses on deriving decorrelated classification weights for the classification head with frozen features, whereas our method emphasizes the improvement of trainable feature embeddings within the feature extractor. Furthermore, there is a distinct difference in the correlation matrices utilized by the two methods. The correlation coefficients

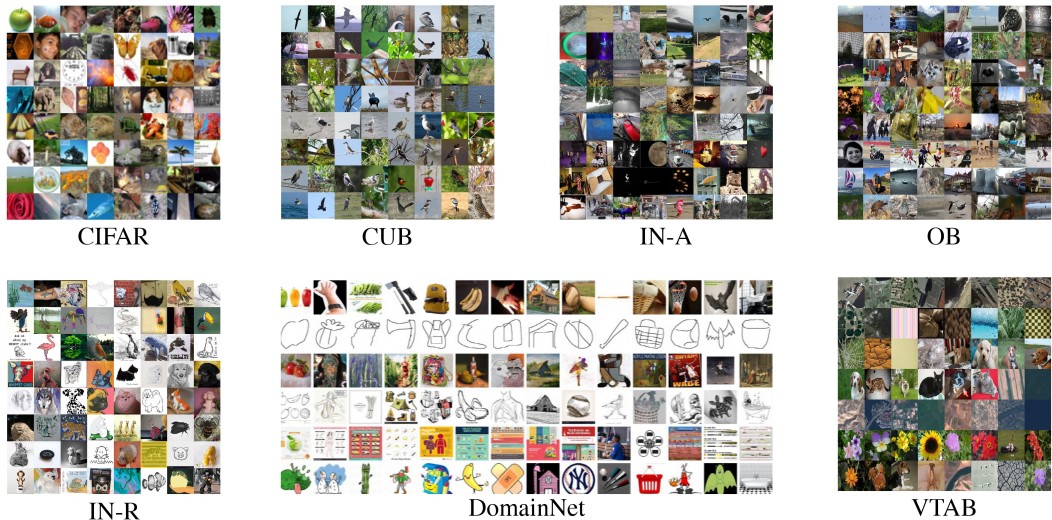

Figure 8: Visualization of seven benchmark datasets.

matrix in RanPAC, as shown in Figure 2 of their paper, has dimensions $\mathbb{R}^{C \times C}$, where $C$ denotes the number of classes in the classification head. In contrast, our method employs a cross-correlation matrix of dimensions $\mathbb{R}^{d \times d}$, with $d$ representing the feature dimension, as detailed in Eq. (1).

In addition, we would like to emphasize that our method is orthogonal to RanPAC. In fact, our approach is built upon RanPAC, and as evidenced in Table 2 of our paper, our method consistently outperforms RanPAC by a significant margin.

# F   Visualization of Datasets

In this section, we provide visualization results of the seven evaluated datasets: CIFAR100 [18], ImageNet-R (IN-R) [11], ImageNet-A (IN-A) [12], CUB200 [38], Omnibenchmark (OB) [50], VTAB [48] and DomainNet [27]. As shown in Fig. 8, SAFE can perform well on datasets with various characteristics. It is noteworthy that SAFE is capable of scenarios where the data distribution between tasks shifts significantly. For example, our method also shows superior performance on VTAB and DomainNet which comprise 5 and 6 distinct tasks, respectively.

