# OpenReview forum: "SAFE: Slow and Fast Parameter-Efficient Tuning for Continual Learning with Pre-Trained Models"
_NeurIPS.cc/2024/Conference — NeurIPS 2024 poster_

### Official Review · Reviewer_RFpE · 2024-07-09

**Soundness:** 2
**Presentation:** 3
**Contribution:** 3
**Rating:** 4
**Confidence:** 5

**Summary:**

This paper proposes a slow and fast parameter-efficient tuning method for continual learning. Slow learner is learned on the first session and fixed with a transfer loss. Fast learner is continually updating for new tasks. Slow learner and Fast learner are further restricted to avoid forgetting.

**Strengths:**

The idea of slow and fast learning has been interesting for continual learning, such as in SLCA. This work leverages pre-trained models to design the slow and fast learner, which is novel. And the experimental results compared to other baselines are significant.

**Weaknesses:**

Although the main idea is sound, it is not well supported by the experimental results. For instance, In Table 2, it shows that with only Fast learner the performance is already as good as the proposed method SAFE. In Table 3, the gain of the aggregation method is marginal compared to some straightforward baselines.  In Table 5, without using $L_{f2s}$ and $L_{s2f}$, the Fast learner obtains the similar results. All these components are claimed as main contributions.

**Questions:**

Can you explain the marginal improvements of some components and how they can support the main contribution of the paper?

**Limitations:**

Yes, it is addressed in the appendix.

---

> ### Author Rebuttal · Authors · 2024-08-07
>
> We will include these experiments and provide detailed explanations of the results in the revised version.
>
>
>
> Q1: The performance with only fast learner (FL) is already as good as the proposed method SAFE, which may not support the main idea.
>
> A1: **First**, to further validate the effectiveness of the slow learner (SL) in SAFE, we present **additional ablation results on six datasets** in Tab. 7 in rebuttal PDF. Notably, **1)** SAFE's final accuracy exceeds that of the FL by an average of 0.4% across datasets. **2)** Specifically, **in ImageNet-R, SAFE surpasses the FL by 0.93%** in final accuracy and **1.55%** in average accuracy.
>
> **Second**, it is crucial to recognize the notable progress in continual learning, particularly given the challenge of balancing stability and plasticity. **For example, EASE [1]** improved final accuracy by **0.03%** over the second-best result and increased average accuracy by **0.08%** on the CUB dataset. On OmniBenchmark, EASE outperformed the second-best result by **0.06%**. Although these improvements might seem minor, EASE has been influential and well-regarded, having been **accepted by CVPR 2024 and cited nine times in three months**. **Similarly, FeCAM** [2], **accepted by NeurIPS 2023**, achieved a **0.2%** final accuracy gain on the ImageNet-Subset, although it **did not surpass recent exemplar-based SOTA** on CIFAR. Compared to these recent SOTAs, **our improvements are significantly more substantial**.
>
> **Third**, we emphasize that the final accuracies of the proposed **SL and FL exceed the second-best** baseline by **3.24% and 4.29%**, respectively, demonstrating **both of them are effective**.
>
>
>
> Q2: The gain of the aggregation method is marginal compared to some straightforward baselines in Table 3. Without using Lf2s and Ls2f, the FL obtains the similar results.
>
> A2: **First**, the **competitiveness** of the proposed aggregation lies in its **ease of implementation and effectiveness**. It achieves **at least a 0.53%** improvement in final accuracy compared with straightforward baselines, **without requiring complex modules or hyperparameters**.
>
> **Second**, the result without Ls2f/f2s represents **a relatively** **enhancement over** Lcos, which is more **challenging**. **Additionally**, when **using only Ls2f/f2s in FL**, the final accuracy increased by **3.1% compared with the baseline**, demonstrating its effectiveness.
>
> **Third**, it is important to note that **this improvement is significant**, especially **in the field of continual learning for PTMs**. For reference, **other markable PTM-based continual learning works** also demonstrate that even **small improvements are noteworthy**.
>
> For instance, **EASE [1]**, which has been **accepted by CVPR 2024** as a notable PTM-based continual learning work, surpassed the second-best result in final accuracy by only **0.03%** and increased the average accuracy by **0.08%** on the CUB dataset. Additionally, EASE surpassed the second-best result by **0.06%** on OmniBenchmark.
>
> **Another example is SSIAT** [2], also **accepted by CVPR 2024**. In experiments on CIFAR, it achieved SOTA results with an increase of **0.09%** in final accuracy and **0.06%** in average accuracy.
>
>
>
> Q3: How each component can support the main contribution of the paper?
>
> A3: **First**, we emphasize that one of the main contributions is **leveraging PTMs to design both SL and FL**.
>
> **Second**, **SL and FL are complementary counterparts and cannot be separated** for the following reasons: **1)** Based on **complementary theory**, **SL** leverages the intrinsic knowledge in PTMs akin to structural knowledge in the **neocortex**, while **FL** continuously learns episodic information for novel classes similar to the **hippocampus**. **2)** As shown in Fig. 7(b) in the additional PDF and Fig.4 in the submitted manuscript, **FL compensates for the plasticity issue of SL** and demonstrates superiority in new classes. **3)** **SL guides the adaptation of FL**, and Tab. 11 in PDF demonstrates that SL is the best teacher for FL.
>
> **Third**, the **knowledge transfer loss functions in SL cannot be considered separately** because the design of two loss functions is **theoretically** **supported by information bottleneck theory (IB)**. We show in Fig.6 in rebuttal PDF that L_diag and L_rdn can be directly related to the corresponding terms in the IB theory, which is used to inherit generalizability and enhance discriminability respectively.
>
> **Fourth**, **the alignment of features**, i.e., L_f2s/s2f and L_cos, **work together** to boost the plasticity of the model without severe forgetting **because** they alleviate the forgetting of the feature extractor and classification head, respectively. **First**, **L_cos** regularizes the training of FL by **aligning features** from FL and SL **in a hypersphere to preserve prior representations**. **Second**, **L_s2f/L_f2s** are two symmetric losses that employ cross-classification to **maintain previous decision boundaries**. For example, the **first component of L_s2f** ensures that the features from FL are compatible with the classification weights of SL. Since our work does not store data or data distributions for replay, in **the second term of L_s2f**, we make sure classification weights from FL (viewed as prototypes for old classes) are correctly classified by SL to mitigate forgetting previous knowledge.
>
>
>
> [1] Expandable subspace ensemble for pre-trained model-based class-incremental learning. CVPR 2024.
>
> [2] Exploiting the heterogeneity of class distributions in exemplar-free continual learning. NeurIPS 2023.
>
> [3] Semantically-Shifted Incremental Adapter-Tuning is A Continual ViTransformer. CVPR 2024.

---

> > ### Comment · Reviewer_RFpE · 2024-08-11
> >
> > Thank you for the responses. However, I remain unconvinced that the improvements are significant, so I will maintain my initial score.

---

> ### Author Response · Authors · 2024-08-11
> **Further Clarifications (1/3)**
>
> Thank you for taking the time to review our previous response and provide your feedback. We appreciate your concerns and would like to further **clarify the effectiveness of our work from the following four perspectives** for your consideration.
>
> **A. Significant Overall Performance of Our Method**
>
> First of all, as validated in Table 1 of our paper, compared to recent SOTAs, **our method shows substantial improvements across all 6 popular datasets**. For example, we surpass the second-best result on ImageNet-A over **4%** which indicates notable superiority.
>
> Second, we would like to further highlight the **significant relative improvements** against previous methods. In particular, we compare our method with two representative SOTA methods, RanPAC [1] and SSIAT [2], from NeurIPS2023 and CVPR2024 respectively, and results are summarized in the table below. We remark that 1) SSIAT [2] did not report (denoted by symbol "-") the result on Omnibenchmark (OB) and hence we compute both the average of all available datasets except OB and the average of all 6 datasets; 2) Metrics in the brackets denotes the corresponding increase (↑) / decrease (↓) compared to its previous SOTA.
>
> | Method               | CIFAR | IN-R        | IN-A        | CUB         |  VTAB      |   OB | Avg of first 5 datasets (Avg5)       | Avg of all 6 datasets (Avg6)       |
> |----------------------|-------|-------------|-------------|-------------|-------------|------|-------------|-------------|
> | RanPAC [1] (NeurIPS 2023) | 92.2  | 78.1        | 61.8        | 90.3        | 92.6        | 79.9 | 83.0        | 82.5        |
> | SSIAT [2] (CVPR 2024)     | 91.4 (↓0.8) | 79.6 (↑1.5) | 62.2 (↑0.4) | 88.8 (↓1.5) | 94.5 (↑1.9) | -    | 83.3 (↑0.3) | -           |
> | **Ours**            | **92.8 (↑1.4)** | **81.0 (↑1.4)** | **66.6 (↑4.4)** | **91.1 (↑2.3)**  | **95.0 (↑0.5)** | **80.9 (↑1.0)** | **85.3 (↑2.0)** | **84.6 (↑2.1)** |
>
> One can observe from the table that:
> - **The relative improvements of our method compare favorably against those of the most recent SOTA SSIAT [2]**. Instead of only 0.3% increase of SSIAT [2] in Avg5 against its previous SOTA RanPAC [1], our method boosts the metric by a significant margin 2.0% compared to SSIAT [2] and outperforms RanPAC [1] by 2.1% in Avg6 metric.
> - Our method shows **consistent improvement over all datasets**. Compared to SSIAT [2] even performs inferior to its previous SOTA RanPAC [1] on CIFAR and CUB, our method outperforms previous methods on all the datasets with various characteristics. This validates the robustness of the proposed method.
>
> Based on the above observations, we believe that our method is sufficiently effective when compared with baselines.
>
> In addition, we would like to remind the reviewer that due to the **inherent difficulty of the tasks**, the trade-off between stability and plasticity, and the fact that our method builds upon pre-trained models (PTMs) rather than starting from scratch, these improvements are even more challenging to achieve. **Other methods show less significant improvements** [3][5]. For instance, **EASE** [5], accepted by **CVPR 2024**, achieved a **0.03%** improvement in final accuracy and a **0.08%** increase in average accuracy on CUB compared to the second-best result. On OmniBenchmark, EASE outperformed the second-best result by **0.06%**. **FeCAM** [3], accepted by **NeurIPS 2023** did **not** surpass recent exemplar-based state-of-the-art results on CIFAR.
>
> [1] RanPAC: Random Projections and Pre-trained Models for Continual Learning. NeurIPS2023.
>
> [2] Semantically-Shifted Incremental Adapter-Tuning is A Continual ViTransformer. CVPR2024.
>
> [3] Exploiting the heterogeneity of class distributions in exemplar-free continual learning. NeurIPS2023.
>
> [4] SLCA: Slow Learner with Classifier Alignment for Continual Learning on a Pre-trained Model
>
> [5] Expandable subspace ensemble for pre-trained model-based class-incremental learning. CVPR2024.
>
> [6] A Unified Continual Learning Framework with General Parameter-Efficient Tuning. ICCV2023.

---

> ### Author Response · Authors · 2024-08-11
> **Further Clarifications (2/3)**
>
> **B. Notable Component-Wise Improvement**
>
> Our method is composed of three key components: a slow learner, a fast learner, and an aggregation module. We would like to emphasize that **each component plays a significant role in the overall effectiveness of our approach**.
>
> 1) **Slow Learner**: Our method demonstrates **significant improvements over the baseline and other alternatives**: **1)** As shown in Table 2 of the manuscript, the proposed slow learner surpasses the baseline method RanPAC [1] by **3.24%** on ImageNet-A. **2)** As indicated in Table 7 of the rebuttal PDF, the proposed slow learner achieves an **average improvement of 1.19%** in final accuracy **across six datasets** compared to the baseline. **3)** As shown in Table 4 of the manuscript, the slow learner **outperforms competing methods** that also aim to transfer statistical knowledge in PTMs. The final accuracy of the slow learner improves by **2.63%** compared to first-order information transfer and by **2.24%** compared to second-order information transfer. **4)** The visualization via t-SNE shows **superior generalization** of the slow learner **on unseen classes** in Figure 3 in the manuscript. The key results are summarized in the following table.
>
> |    Method    | Performance (%) |            Competitor             | Data Source |
> |:------------:|:---------------:|:---------------------------------:|:-----------:|
> | Slow learner |      +3.24      |             Baseline              |   Table 2   |
> | Slow learner |      +2.63      | First-order information transfer  |   Table 4   |
> | Slow learner |      +2.24      | Second-order information transfer |   Table 4   |
>
> 2) **Fast Learner**: Our method shows **notable improvements compared to the baseline and other approaches**. Specifically: **1)** As shown in Table 2 of the manuscript, the proposed fast learner **surpasses the baseline by 4.29%** on ImageNet-A. **2)** As indicated in Table 7 of the submitted PDF, the proposed fast learner achieves an **average improvement of 1.72%** in final accuracy **across six datasets** compared to the baseline. **3)** The following ablation table, which extends Table 5 of the manuscript with a new row, demonstrates that **both $\mathcal{L}\_{\mathrm{cos}}$, $\mathcal{L}\_{\mathrm{f2s/s2f}}$ contribute significant improvements.** Specifically, solely using **$\mathcal{L}\_{\mathrm{f2s/s2f}}$** results in an improvement of **3.10%** compared to the baseline, while **$\mathcal{L}\_{\mathrm{cos}}$** yields a gain of **3.86%** over the baseline. **4)** As shown in Table 11 of the rebuttal PDF, the fast learner guided by the slow learner achieves an improvement in final accuracy of **10.73%** and **2.83%** compared to **other teacher choices**. **5)** As shown in Figure 7(b) in the additional PDF and Figure 4 in the submitted manuscript, **fast learner compensates for the plasticity** issue of slow learner and demonstrates superiority in new classes.
>
> | Method                                                                            | Final     | Avg       |
> | --------------------------------------------------------------------------------- | --------- | --------- |
> | Baseline                                                                          | 62.21     | 72.31     |
> | Fast Learner w/o $\mathcal{L}\_{\mathrm{f2s/s2f}}$, $\mathcal{L}\_{\mathrm{cos}}$ | 8.16      | 30.73     |
> | Fast Learner w/o $\mathcal{L}\_{\mathrm{cos}}$                                     | 65.31     | 73.88     |
> | Fast Learner w/o $\mathcal{L}\_{\mathrm{f2s/s2f}}$                                 | 66.07     | 74.20     |
> | **Fast Learner**                                                                  | **66.49** | **74.50** |
>
>
> 3) **Aggregation Module**:
> - **First**, the proposed aggregation method demonstrates **significant improvements over the baseline and other alternatives**. Specifically: **1)** As indicated in Table 7 of the rebuttal PDF, the proposed aggregation approach achieves an **average improvement of 2.1%** in final accuracy **across six datasets** compared to the baseline. **2)** As shown in Table 3 of the submitted manuscript, the aggregation method results in **at least a 0.53% improvement** in final accuracy compared to straightforward baselines.
> - **Second**, the proposed aggregation method **does not require complex modules or trainable parameters**, making it straightforward to implement.
> - **Third**, it is important to recognize that **aggregate** methods are generally more **challenging to improve**. For example, the **LAE**[6] method from **ICCV 2023** achieved only a **0.4%** increase in average accuracy, with a **0.4% decrease** in final accuracy on ImageNet-R. In contrast, our method demonstrates a **2.85% final** improvement on ImageNet-R and an **average** improvement of **2.26%**.

---

> ### Author Response · Authors · 2024-08-11
> **Further Clarifications (3/3)**
>
> **C. Unified Framework Synergy**
>
> We would like to emphasize that our approach serves as **a unified framework** for PTM-based continual learning. Specifically, we would like to clarify that:
>
> 1) The effectiveness of our framework lies not only in the contribution of each individual component but also in the overall improvement it delivers, which can make the **impact of each component less apparent** when viewed in isolation. A similar situation is **observed in RanPAC** [1], accepted by **NeurIPS 2023**, where Phase 1 (a single component) results in improvements of only **0.3%** on CUB and OmniBenchmark, and **0.2%** on VTAB. In contrast, our two learners achieve a minimum improvement of **0.36%** on CUB, **0.75%** on OmniBenchmark, and **1.11%** on VTAB. **Furthermore**, our method consistently demonstrates **substantial improvements across all six popular datasets**.
>
> 2) The **components** of our method are **interdependent** and should not be separated. For example, the **fast learner relies on the slow learner for guidance** during adaptation. As demonstrated in Table 11 of the rebuttal PDF, the performance of the fast learner degrades by **10.73%** and **2.83%** when guided by alternative teachers.
>
>
>
> **D. Contributions Beyond Experimental Results**
>
> We would like to draw the reviewer's attention to the **unique contributions and features** of our work (particularly the **theoretical** foundations that set our work apart):
>
> 1) To the best of our knowledge, we are the first to apply **slow and fast learning to parameter-efficient tuning** for addressing challenges in PTM-based continual learning.
> 2) Our slow learner inherits generalizable knowledge from PTMs, supported by **theoretical analysis** using information bottleneck theory, a crucial aspect often **overlooked in previous works**. This transfer enables the slow learner to generalize well to novel classes that it has not encountered during training.
> 3) We address the **current limitations in adaptation** by guiding the fast learner with the slow learner, **eliminating the need for access to data or data distributions** [2][4]. During this process, the proposed fast learner continuously adapts to new knowledge while effectively mitigating the forgetting of previous knowledge.
>
> ---
>
> In conclusion, we believe that the thorough analysis and evidence provided in our response demonstrate the robustness and effectiveness of our proposed method. By addressing each of the reviewer's concerns and presenting detailed comparisons with SOTA approaches, we have shown that our work not only advances the field but also offers unique contributions through its theoretical foundations and innovative components. **We sincerely hope that this additional clarification helps to illustrate the significance of our research and leads to a favorable evaluation.** Thank you once again for your thoughtful review and consideration.

---

### Official Review · Reviewer_Psm6 · 2024-07-11

**Soundness:** 2
**Presentation:** 3
**Contribution:** 2
**Rating:** 4
**Confidence:** 5

**Summary:**

The paper proposes a novel method named SAFE (Slow And Fast parameter-Efficient tuning) to tackle challenges in continual learning. SAFE introduces a unified framework that combines slow parameter-efficient tuning (S-PET) for inheriting general knowledge from pre-trained models (PTMs) and fast parameter-efficient tuning (F-PET) for acquiring task-specific knowledge in incremental sessions. SAFE demonstrates state-of-the-art performance on six benchmark datasets, showing significant improvements over existing methods.

**Strengths:**

1. SAFE introduces an innovative approach by integrating both slow and fast parameter-efficient tuning within a unified framework. This dual-tuning mechanism effectively addresses the trade-off between stability and plasticity in continual learning.
2. The paper presents a thorough and well-structured evaluation on six benchmark datasets, clearly demonstrating the effectiveness and superiority of SAFE over existing methods.
3. The approach does not rely on storage class distributions for data replay and maintains constant computational and memory overheads.

**Weaknesses:**

1. Training the slow adapter only in the initial session may restrict its ability to adapt to new tasks, potentially limiting the overall flexibility of the model.
2. Complex Loss Functions and Hyperparameters: The proposed method involves multiple loss functions and hyperparameters, making the optimization process complex and potentially challenging to tune in practice.
3. While the method shows promising results, additional experiments or ablations specifically addressing the impact of each loss function and hyperparameter choice would strengthen the validation of the approach.

**Questions:**

1. Why did you decide to train the slow adapter only in the initial session? Would there be any benefit to periodically updating it with new data to improve adaptability?
2. How did you determine the values for hyperparameters such as λ, Ldiag, Lrdn, and Lcos? Could you provide more details on the process and any automated search methods used?
3. Can you elaborate on the specific contributions of each loss function (knowledge transfer loss and feature alignment loss) in the overall performance of SAFE? How do these losses interact, and are there any potential conflicts between them?
4. Have you considered comparing SAFE with other PET methods that address the stability-plasticity trade-off, such as O-LoRA [1]?
[1] O-LoRA: Orthogonal Subspace Learning for Language Model Continual Learning, CVPR

**Limitations:**

The authors acknowledge that SAFE's training process involves complex loss functions and hyperparameters, which could be a barrier to practical implementation.

---

> ### Author Rebuttal · Authors · 2024-08-07
>
> Thanks for your suggestions. We will include the mentioned experiments and detailed explanations in the revised version.
>
>
>
> Q1: Training the slow learner (SL) only in the initial session may restrict its ability to adapt to new tasks, potentially limiting the overall flexibility of the model. Why did you decide to train the SL only in the initial session? Would there be any benefit to periodically updating it with new data to improve adaptability?
>
> A1: **First**, **training SL only in the initial session does not impede the overall flexibility** because: **1)** It is a **common practice in incremental learning** [1][2][3] to **develop a generalizable model capable of being transferred to diverse subsequent tasks**. **2)** The **SL** inherits the generalizability of PTMs, providing the **flexibility to adapt to unseen categories** in the future. This is **verified through: (a)** Experiment results show SL **surpasses the baseline by 3.23%** in Tab. 3 in manuscript and performs will on DomainNet in Tab. 8 in PDF. **(b)** Visualization via t-SNE shows **better clustering results on unseen classes** in Fig. 3 in manuscript **3)** In alignment with **complementary theory**, Fig. 7(b) in PDF demonstrates the **adaptation complementary of SL by FL**, as the latter quickly adapts across sessions.
>
> **Second**, we decided to **train SL only in the initial session rather than periodically updating** because: **1)** Continuously updating may **lead to the forgetting of general knowledge** **inherited** from PTMs. **2)** The performance of the **FL** may **deteriorate without the guidance of frozen SL** as shown in Tab. 11 in PDF. **3)** We **have experimented** with updating SL using EMA or a small learning rate, but **observed no noticeable improvement**. We will address this problem in future work.
>
>
>
> Q2: The proposed method involves multiple loss functions and hyperparameters. How did you determine the values for hyperparameters? Could you provide more details on the process and any automated search methods used?
>
> A2: **First**, only **4 additional loss functions** accompanied with **4 hyperparameters** are used, maintaining reasonable complexity. We have **provided a set of valid default parameters** for implementation (line 246 in manuscript).
>
> **Second**, we provide the **grid search results** in PDF: the results for λdiag and λrdn of SL of Fig. 7(a) in PDF, for λcos of FL in Tab. 9, and for *γ* of aggregation in Tab. 10.
>
> **Third**, the **results remain relatively stable** within a certain range, making a **default set of hyperparameters effective** and simplifying the parameter tuning process. For example, SL can achieve satisfactory results with λdiag in 0.1-1 and λrdn in 100-500. FL can obtain good performance with λcos in 50-100. The aggregation module works well by simply setting γ to 1.
>
>
>
> Q3: Additional experiments or ablations specifically addressing the impact of each loss function and hyperparameter choice would strengthen the validation of the approach.
>
> A3: **First,** we **add the following experimental results** in PDF to verify **components**: **1)** Tab. 7 to demonstrate the significant performance improvements across various datasets, elucidating the necessity of each module. **2)** Fig. 7(b) illustrates how the aggregation mechanism effectively leverages the complementary strengths of SL and FL. **3)** Tab. 8 shows the superiority of SAFE on dataset with domain gap (DomainNet).
>
> **Second**, regarding **hyperparameter** selection: **1)** The grid search results are provided in PDF: Fig. 7(a) displays the results for λdiag and λrdn of SL, Tab. 9 presents the results for λcos of SL, and Tab. 10 shows the results for *γ*. **2)** The results indicate that **a default set of hyperparameters (line 246) is practically effective, as they remain relatively stable within a certain range**.
>
>
>
> Q4: Elaborate on the specific contributions of each loss function in the overall performance of SAFE. How do these losses interact, and are there any potential conflicts between them?
>
> A4: **First,** based on **complementary theory**, **SL and FL are complementary counterparts**, **further substantiated** by Fig. 7(b) in rebuttal PDF and Fig.4 in manuscript.
>
> **Second**, the knowledge transfer **loss functions in SL** work collaboratively to **facilitate the effective transfer of general knowledge from PTM**, which is **proposed for the first time**. **Moreover,** the designs of Ldiag and Lrdn in SL are **theoretically** supported by information bottleneck theory (IB). We show in Fig.6 in rebuttal PDF that Ldiag and Lrdn can be directly related to the corresponding terms in the IB theory.
>
> **Third,** Lf2s/s2f and Lcos in FL **work together** to boost the plasticity **without storing data or data distributions for replay**. **Lcos** regularizes the training of FL by **aligning features** from FL and SL **in a hypersphere to preserve prior representations**. **Ls2f/Lf2s** are two symmetric losses that employ cross-classification to **maintain previous decision boundaries**.
>
>
>
> Q5: Have you considered comparing SAFE with other PET methods that address the stability-plasticity trade-off, such as O-LoRA.
>
> A5: We **did not** **compare** SAFE with O-LoRA due to the following reasons: **1)** They are designed for **different application domains**: O-LoRA is specifically tailored for language models, while SAFE is focused on computer vision. **2)** Comparing **disparate PET methods**, such as LoRA and Adapter, may not be entirely equitable. However, **SAFE has demonstrated robustness and generality** across various PET approaches. We are willing to **explore the extension** of SAFE in future work.
>
>
>
> [1] Forward compatible few-shot class-incremental learning. CVPR2022.
>
> [2] First session adaptation: A strong replay-free baseline for class-incremental learning. CVPR2023.
>
> [3] Semantically-Shifted Incremental Adapter-Tuning is A Continual ViTransformer. CVPR2024.

---

> > ### Comment · Reviewer_Psm6 · 2024-08-12
> > **Official Comment by Reviewer Psm6**
> >
> > Many thanks to the authors for the detailed answers.
> > After considering both the other reviews and the rebuttals, I maintain the original rating.

---

> ### Author Response · Authors · 2024-08-12
> **Further Clarifications**
>
> Thank you for your feedback. We greatly appreciate the time and effort you have put into reviewing our submission. We have carefully considered your concerns and have endeavored to address them in our rebuttal. Below, we would like to summarize the key points regarding your raised concerns.
>
> 1) Regarding the concerns about training the slow adapter only in the initial session, we have provided both qualitative and quantitative explanations to demonstrate its effectiveness and necessity compared to periodic updates.
> 2) For concerns related to loss functions and hyper-parameters, we offer detailed ablations and analyses to highlight their specific contributions and to show the stability of the default hyper-parameter choices, which facilitate practical implementation.
> 3) Additionally, we provided a specific explanation in the manuscript for why the proposed method was not compared with O-LoRA, an inspiring work in continual learning. Thank you for bringing this to our attention; **we will include O-LoRA** in the Related Work section of the **revised version** for discussion and comparison.
>
> Moreover, we would like to draw the reviewer's attention to the **unique contributions and features** of our work, particularly the **theoretical foundations** that set our work apart.
> 1. To the best of our knowledge, we are the first to apply **slow and fast learning to parameter-efficient tuning** in addressing challenges in PTM-based continual learning.
> 2. Our slow learner inherits generalizable knowledge from PTMs, underpinned by **theoretical analysis** using information bottleneck theory—a crucial aspect often **overlooked in previous works**. This transfer allows the slow learner to generalize effectively to novel classes it has not encountered during training.
> 3. We tackle the current limitations in adaptation by guiding the fast learner with the slow learner, thereby **eliminating the need for access to data or data distributions**. Throughout this process, the proposed fast learner continuously adapts to new knowledge while effectively mitigating the forgetting of previous knowledge.
> 4. The clear demonstration of the effectiveness and superiority of our proposed method.
>
> We hope these additional details and clarifications address your concerns. **We welcome any further discussion** to enhance the clarity and quality of our work. Considering the extensive elaborations and the unique contributions of our research, **we would like to humbly request a reconsideration of the scoring**.
>
> We believe our research offers valuable insights and contributions that would be of significant interest to the NeurIPS community.
>
> Thank you again for your time and thoughtful consideration.

---

### Official Review · Reviewer_JgdG · 2024-07-13

**Soundness:** 3
**Presentation:** 3
**Contribution:** 2
**Rating:** 4
**Confidence:** 5

**Summary:**

This paper proposes a novel paradigm for continual learning called SAFE, which utilizes both slow and fast parameter updates. The method focuses on continual learning with pre-trained models, using slow updates to preserve the generalization capability of the pre-trained model while employing fast updates to adapt to new downstream tasks. During inference, the two branches are aggregated to achieve more robust predictions. Experiments on 6 different benchmarks demonstrate that SAFE consistently outperforms existing methods, achieving state-of-the-art performance.

**Strengths:**

1. The paper focuses on continual learning with pre-trained models, a highly valuable topic. By designing a slow learner, it retains the generalization capability of the pre-trained model.
2. The paper also designs a fast learner to balance stability and plasticity, enabling the model to continually adapt to new downstream tasks.
3. The paper conducts extensive experiments on multiple benchmarks, providing strong evidence for the effectiveness of the proposed method.

**Weaknesses:**

1. The slow learner assumes that the data distribution of the first task in the continual learning scenario is roughly similar to subsequent tasks, which is not always true in practical applications. When the data distribution of the first task diverges significantly from subsequent tasks, constraining the fast learner to the slow learner may yield negative results.
2. The design of the method introduces too many loss function terms, which could make the model training overly complex. Additionally, the stability of these hyperparameters is not analyzed in the experimental section.
3. The main experimental section lacks comparisons with the latest methods, such as DAP[1].
4. The ablation studies show that the introduction of many components results in only marginal improvements. According to Occam's Razor, these components should be removed.

[1] Jung D, Han D, Bang J, et al. Generating instance-level prompts for rehearsal-free continual learning[C]. Proceedings of the IEEE/CVF International Conference on Computer Vision. 2023: 11847-11857.

**Questions:**

1. I am curious about SAFE's performance in scenarios where the data distribution between tasks varies significantly, such as in DomainNet.
2. What is the setup for the ablation study in Table 2? Does "only the fast learner" mean that only the SL is used for inference after training, or does it mean that only SL is involved in the training process?
3. What is the setup for PET in SAFE? Does it include only one of Adapter, SSF, or VPT, or does it use all three simultaneously?

**Limitations:**

The method proposed in the paper is only applicable to scenarios where the data distribution of the first task is roughly similar to that of subsequent tasks.

---

> ### Author Rebuttal · Authors · 2024-08-07
>
> Thanks for your suggestions. We will include the mentioned experiments and analysis in the revised version.
>
> Q1: The slow learner (SL) assumes that the data distribution of the first task in the continual learning (IL) scenario is roughly similar to subsequent tasks, which is not always true in practical applications. When the data distribution of the first task diverges significantly from subsequent tasks, constraining the fast learner (FL) to the SL may yield negative results.
>
> A1: **First, SL does not have the assumption.** Instead, SL is adapted solely in the first session with the aim of developing a generalizable model capable of being transferred to diverse subsequent tasks [1][2][3].
>
> **Second, constraining FL to SL does not yield negative results both theoretically and experimentally**. SL inherits generalizability from PTMs, which is **theoretically supported by information bottleneck theory** (Fig. 6 in PDF). **Experimentally**, **SAFE achieves SOTA on datasets where the data distribution of the first task diverges significantly from subsequent tasks**, such as VTAB (Tab. 1 in manuscript) and DomainNet (Tab. 7 in PDF). VTAB includes 19 assessment tasks across a variety of domains, while DomainNet features 6 common objects in different domains.
>
> **Third, we find that not constraining FL to SL yields worse results.** As shown in Tab. 11 in PDF, FL guided by frozen SL achieves SOTA.
>
> Q2: The design of the method introduces too many loss function terms, which could make the model training overly complex. Additionally, the stability of these hyperparameters is not analyzed.
>
> A1: **First**, only 4 additional loss functions accompanied with 4 hyperparameters are used in SAFE, maintaining a **reasonable level of complexity**.
>
> **Second**, we provide the **grid search results** in PDF: the results for λdiag and λrdn in Fig. 7(a), for λcos in Tab. 9, and for *γ* in Tab. 10. The **results remain relatively stable within a certain range, making the default set of hyperparameters** (Line 246 in manuscript) **effective** and simplifying the parameter tuning process.
>
> Q3: The main experimental section lacks comparisons with the latest methods, such as DAP.
>
> A3: **We did not compare SAFE with DAP because** DAP involves direct annotation of task identity, simplifying the difficulty during inference [4], which renders such comparisons unfair. We now present the **comparisons with DAP** in Tab. 12 in PDF and find **SAFE surpasses DAP across datasets**.
>
> Q4: The ablation studies show that the introduction of many components results in only marginal improvements.
>
> A4: **The improvement is not marginal in IL**. **First,** the improvement of 0.x% in **average accuracy** **reflects a** **consistent increase** of 0.x% in the accuracy of each session, which is a noteworthy enhancement. **Second**, other works in PTM-based IL also **highlight that the modest improvements are impactful.** For instance, In EASE[5], a notable PTM-based IL work accepted at CVPR2024, the final acc exceeded the competitor by 0.03%, with the average acc increasing by 0.08% on CUB. Additionally, EASE surpassed the second-best result by 0.06% on OmniBenchmark. Another example is SSIAT[3], which was also accepted by CVPR 2024. In experiments on CIFAR, SSIAT achieved SOTA with an increase of 0.09% in final and 0.06% in average acc.
>
> **These components should not be removed, not only because of the performance improvement but collaboration**: 1) Based on **complementary theory**, SL leverages the intrinsic knowledge in PTMs akin to structural knowledge in **neocortex**, while FL continuously learns episodic information for novel classes similar to the **hippocampus**. 2) As shown in Fig. 7(b) in PDF and Fig.4 in manuscript, **FL compensates for the plasticity issue of SL** and shows superiority in new classes. 3) **SL guides the adaptation of FL**, and Tab. 11 in PDF demonstrates that SL is the best teacher for FL. 4) The designs of Ldiag and Lrdn in SL are **theoretically** supported by information bottleneck theory (IB). As Fig.6 in rebuttal PDF shows, Ldiag and Lrdn can be directly related to the corresponding terms in the IB theory. 5) Lf2s/s2f and Lcos in FL **work together** to boosts the plasticity. Lcos regularizes the training of FL by aligning features from FL and SL in a hypersphere to preserve prior representations. Ls2f/Lf2s are two symmetric losses that employ cross-classification to **maintain previous decision boundaries**.
>
> Q5: SAFE's performance in scenarios where the data distribution between tasks varies significantly e.g.DomainNet.
>
> A5: **SAFE is applied to datasets with significantly varying data distributions between tasks**, such as VTAB and DomainNet (Tab. 8 in PDF), and achieves SOTA. VTAB comprises 5 assessment tasks spanning various domains, including natural, professional, and structured images.
>
> Q6: Does only the fast learner in Table 2 mean only SL is used for inference after training?
>
> A6: It means only the slow learner is used for inference after training.
>
> Q7: Does the setup for PET in SAFE include only one of Adapter, SSF, or VPT, or does it use all three simultaneously?
>
> A7: First, following previous work [5][6], we employ only one PET in SAFE for the experiments, and the specific PET used is consistent with that in [5].
>
> Second, we validate in Tab. 6 that SAFE is compatible with various PET methods, including Adapter, SSF, and VPT, thereby demonstrating its generalizability.
>
> [1] Forward compatible few-shot class-incremental learning. CVPR2022.
>
> [2] First session adaptation: A strong replay-free baseline for class-incremental learning. CVPR2023.
>
> [3] Semantically-Shifted Incremental Adapter-Tuning is A Continual ViTransformer. CVPR2024.
>
> [4] Continual learning with pre-trained models: A survey. ArXiv2024.
>
> [5] Expandable subspace ensemble for pre-trained model-based class-incremental learning. CVPR2024.
>
> [6] Ranpac: Random projections and pre-trained models for continual learning. NeurIPS2023.

---

> > ### Comment · Reviewer_JgdG · 2024-08-12
> >
> > Thanks for your detailed responses. However, regarding the concerns I raised, I still remain unconvinced. After considering the opinions of other reviewers and the authors' rebuttal, I have decided to maintain my initial score.

---

> ### Author Response · Authors · 2024-08-12
> **Further Clarifications**
>
> Thank you for your thoughtful feedback. We greatly appreciate the time and effort you have put into reviewing our submission. We have carefully considered your concerns and have endeavored to address them in our rebuttal. Below, we would like to summarize the key points regarding your raised concerns.
>
> 1. We provide further explanations to address your misunderstanding that the slow learner operates under certain assumptions. We also experimentally verified that constraining the fast learner to the slow learner yields better results.
> 2. We apply the proposed method to datasets with significantly varying data distributions between tasks, such as VTAB and DomainNet, and achieve state-of-the-art results. This demonstrates that our method is effective in scenarios where the data distribution of the first task diverges significantly from that of subsequent tasks.
> 3. For the concerns related to loss functions and hyper-parameters, we offer additional ablations and details to highlight the stability of the default choices, which are straightforward to implement in practice.
> 4. Thank you for bringing DAP to our attention. **We will include a discussion and comparison with it in the Related Work** of the revised version, further demonstrating the superiority of our approach.
> 5. Regarding concerns about the improvements of each component, we provide additional experimental results to demonstrate the significant improvement of each component in the overall effectiveness of our approach and their collaborative impact.
>
> In addition, we would like to draw the reviewer's attention to the **unique contributions and features** of our work, particularly the **theoretical foundations** that distinguish it.
> 1.  To the best of our knowledge, we are the first to apply **slow and fast learning to parameter-efficient tuning** in addressing challenges in PTM-based continual learning.
> 2.  Our slow learner inherits generalizable knowledge from PTMs, underpinned by **theoretical analysis** using information bottleneck theory—a crucial aspect often **overlooked in previous works**. This transfer allows the slow learner to generalize effectively to novel classes it has not encountered during training.
> 3.  We tackle the current limitations in adaptation by guiding the fast learner with the slow learner, thereby **eliminating the need for access to data or data distributions**. Throughout this process, the proposed fast learner continuously adapts to new knowledge while effectively mitigating the forgetting of previous knowledge.
> 4. We provide extensive experiments on multiple benchmarks, providing strong evidence for the effectiveness of the proposed method.
>
> We hope these additional details and clarifications address your concerns. **We remain open to further discussion** to ensure the clarity and quality of our work. Given the extensive elaborations and the unique contributions of our research, **we respectfully request a reconsideration of the score**. We believe our work offers valuable insights that would be of great interest to the NeurIPS community.
>
> Thank you again for your time and consideration.

---

### Official Review · Reviewer_bXaj · 2024-07-15

**Soundness:** 3
**Presentation:** 3
**Contribution:** 3
**Rating:** 5
**Confidence:** 5

**Summary:**

The paper introduces the SAFE (Slow And Fast parameter-Efficient tuning) framework for continual learning using pre-trained models (PTMs). The proposed approach combines slow parameter-efficient tuning (S-PET) to inherit general knowledge from PTMs and fast parameter-efficient tuning (F-PET) to adapt to new tasks in each incremental session. The paper validates the effectiveness of the SAFE framework through experiments on multiple datasets, demonstrating its superiority over state-of-the-art methods.

**Strengths:**

- The paper is generally well-organized and easy to follow.
- The method is well motivated and contains clear novelty comparing previous methods.
- The paper provides extensive experimental validation on various benchmark datasets, showing improvements over state-of-the-art methods.

**Weaknesses:**

- The motivation and effects of the proposed techniques are unclear.
    - For example, what are the specific effects of the cross-correlation matrix? What is the specific motivation? This design is quite similar to the de-correlation operation process in RanPAC, although with a different formulation. More solid analyses are required.
    - The issue is still with the fast learner part. Although ablation studies are conducted for the components, the insights of the designs are unclear. For ablation studies, not only the terms should be ablated, but the designs, such as the usage of W_slow in the loss should be clearly discussed and analyzed.
- Although the paper is generally well written, some parts have tedious descriptions and blurry presentations, such as line 58 - 81 in the introduction, which need to be improved.
- The proposed method contains many hyper-parameters, such as the weights for the loss terms. Ablation studies for them are necessary and required.
- The importance of aggregation for the performance should be discussed based on the ablation study.
- The aggregation process requires weights for slow and fast results. Although they can be calculated automatically, it should be analyzed and demonstrated how the distribution of these weights for different data samples and datasets.

The paper can be a good paper. However many details are absent in the papers, making the delivered message unclear and the techniques not justified well. I will reconsider my score based on the rebuttal.

**Questions:**

The questions are left with weakness points,

**Limitations:**

The paper includes a limitation section in the appendix. But it does not cover actual limitations of the work. Instead, the discussed limitation is mainly the characteristics of general pre-trained model-based continual learning (i.e., using the pre-trained model), which is actually not a limitation. The authors may discuss the limitation related to the robustness of the hyper-parameters.

---

> ### Author Rebuttal · Authors · 2024-08-07
>
> Thanks for your comments. We will include these experiments and provide detailed analysis in the revised version.
>
> Q1: What are the specific effects of the cross-correlation matrix and its specific motivation? This design is quite similar to the de-correlation operation process in RanPAC.
>
> A1: The **motivation** behind the cross-correlation matrix 𝑀 is **twofold**. **First**, it aims to create loss functions that **facilitate the effective transfer of general knowledge from PTM** to the slow learner (SL), which is often overlooked in previous works. This transfer enables the SL to generalize well to novel classes that it has not encountered during training. **Second**, M allows for **theoretical analysis** using the Information Bottleneck (IB) theory [1], which can be represented as IB = I(f, θ) − βI(f,X), where X, f and θ denote images, features and model parameters. We show in Fig. 6 in PDF that loss functions derived from 𝑀 can be directly related to the corresponding terms in the IB theory.
>
> The **effects of M are twofold** and its **superiority over other alternatives is confirmed in Tab. 4** in manuscript. **1)** By maximizing the diagonal elements, embeddings from PTM and SL become similar, making **SL inherit generalizability** from PTM. **2)** By minimizing the off-diagonal ones, output units of embedding contain **non-redundant information to enhance discriminability**.
>
> Our design is **different from RanPAC**. **1)** SL uses trainable features with M but RanPAC uses frozen features. **2)** SL employs M during feature training while de-correlation is operated on classification weights. **3)** 𝑀 incorporates PTM into the training process. **4)** Both SL and SAFE consistently outperform RanPAC.
>
> Q2: The insights of the designs of fast learner (FL) are unclear. For ablation studies, not only the terms but the designs, such as the usage of W_slow in the loss should be discussed and analyzed.
>
> A2: The **loss function of the FL** **comprises** **two components**: L_cos and L_s2f/L_f2s, which alleviates the forgetting of the feature extractor and classification head, respectively. **First**, **L_cos** regularizes the training of FL by **aligning features** from FL and SL **in a hypersphere to preserve prior representations**. **Second**, **L_s2f/L_f2s** are two symmetric losses that employ cross-classification to **maintain previous decision boundaries**. For example, the **first component of L_s2f** ensures that the features from FL are compatible with the classification weights of SL (W_slow). Since our work does not store data or data distributions for replay, in **the second term of L_s2f**, we make sure classification weights from FL (viewed as prototypes for old classes) are correctly classified by W_slow to mitigate forgetting previous knowledge.
>
> Q3: Some parts have tedious descriptions and blurry presentations, such as line 58-81 in the introduction.
>
> A3: We aimed to provide readers with comprehensive understanding of SAFE. We will enhance the clarity in not only lines 58-81 but the rest of the paper.
>
> Q4: Ablation studies for hyper-parameters in SAFE are required.
>
> A4: SAFE includes 4 key hyper-parameters: λ_diag and λ_rdn for SL, λ_cos for FL and *γ* for aggregation. We provide a reasonable set of default choices in line 246 of our manuscript.
>
> In PDF, we supply **detailed hyper-parameter sensitivity analyses**. Results for λ_diag and λ_rdn are presented in Fig. 7(a), while the results for λ_cos shown in Tab. 9. Experiments for *γ* are in Tab. 10. It is observed that **hyperparameters remains relatively stable within a certain range**. For example, SL can achieve satisfactory results with λ_diag in 0.1-1 and λ_rdn in 100-500. FL can obtain good performance with λ_cos in 50-100. The aggregation module works well by simply set *γ* to 1.
>
> Q5: The importance of aggregation for the performance should be discussed.
>
> A5: In manuscript, we **validate the effectiveness of aggregation in Tab. 2**, which shows that the **aggregated model achieves both the best** final and average accuracy on IN-A.
>
> To better illustrate its necessity, **experiments on 6 datasets are provided**. As shown in Tab. 7, the final accuracy of the **aggregated model** obtains **0.9% improvement** on IN-R and obtains **0.4% over 6 datasets**.
>
> Q6: The aggregation process requires weights for SL and FL. Although they can be calculated automatically, it should be analyzed and demonstrated how the distribution of these weights for different data samples and datasets.
>
> A6: **First**, **Fig. 7(b) in PDF shows the (softmaxed) average weights** of SL and SL. For test samples from session 1 to session 10, weights of FL increase monotonously while those for SL decrease. This indicates that **FL is more confident in newly learned samples while SL prefers historical ones**, which is consistent with our hypothesis that FL can capture short-term information and SL focuses on long-term structured knowledge.
>
> **Second**, **in combination with the results in Fig 4**, FL and SL primarily exhibits higher weights in sessions 7-10 and 1-6 respectively, where accuracy is higher. This indicates that the **aggregated model can dynamically leverage the strengths of both learners**, validating the improvements observed in the experiments.
>
> Q7: The authors may discuss the limitation related to the robustness of the hyper-parameters.
>
> A7: Our method introduces 3 hyper-parameters to balance loss functions in training. **Although** we find in experiments that **a set of default choices is suitable for 6 datasets** in manuscript, **it may become suboptimal when tested on datasets with significantly different statistical characteristics**. Promising future directions include designing effective hyper-parameter searching mechanisms or generating them based on the relation between PTM pre-trained datasets and downstream task data.
>
> [1] The information bottleneck method. arXiv physics 2000.

---

> ### Comment · Reviewer_bXaj · 2024-08-12
>
> I appreciate the authors’ response. The authors addressed my concerns partially; and I can see the contributions in the paper. I would like to be on the side of acceptance and hope that more detailed analyses and more insightful discussions can be added to improve the paper's quality. Based on the current rebuttal, I would like to maintain my original score.
>
>
> - The authors may further highlight and clarify the relationship with RanPAC for the correlation matrix part. Furthermore, if the matrix is learnable, I am not confident about (or at least it is not clarified or analyzed) what kind of correlation can be learned and how to make sure a valid correlation can be learned, especially considering that the matrix and other parameters are learned simultaneously. I suggest some strong analysis and justification for this part.
>
> - For the ablation studies on aggregation, I want to see how important aggregation inference is and how’s the sensitivity of it, for an already-trained model. It seems Table 2 is the ablation studies of whether slow or fast is included in the whole model, mainly the training process.
>
> - The authors provide a new analysis on the aggregation weights in rebuttal - “in combination with the results in Fig 4, FL and SL primarily exhibits higher weights in sessions 7-10 and 1-6 respectively”. How to explain this observation?
>
> - About the limitation - my comments are only about discussing more essential limitations in the paper.

---

> > ### Author Response · Authors · 2024-08-13
> > **Further Clarifications (1/2)**
> >
> > Thank you for your valuable feedback. We are pleased to note that some of your concerns have been addressed in our previous response and are grateful for your acknowledgment of our contributions. Regarding the remaining issues and suggestions you mentioned, we would like to provide a more thorough discussion in this response. All the analysis, explanations and modifications provided in this response will be included in the revised version of our manuscript.
> >
> > Q1: The authors may further highlight and clarify the relationship with RanPAC for the correlation matrix part.
> >
> > A1: Thank you for your thoughtful suggestion. We appreciate the opportunity to clarify the relationship between our method and RanPAC [1]. While both approaches leverage pre-trained models (PTMs) for continual learning, they target different components of the model. Specifically, RanPAC [1] focuses on deriving decorrelated classification weights for the **classification head** with **frozen features**, whereas our method emphasizes the improvement of **trainable feature embeddings** within the **feature extractor**.
> >
> > Furthermore, there is a distinct difference in the correlation matrices utilized by the two methods. The correlation coefficients matrix in RanPAC [1], as shown in Figure 2 of their paper, has **dimensions $\mathbb{R}^{C \times C}$**, where **$C$ denotes the number of classes** in the classification head. In contrast, our method employs a cross-correlation matrix $\boldsymbol{M}$ of **dimensions $\mathbb{R}^{d \times d}$**, with **$d$ representing the feature dimension**, as detailed in line 156 of our manuscript.
> >
> > Moreover, we would like to emphasize that **our method is orthogonal to RanPAC** [1]. In fact, our approach builds upon RanPAC, and as evidenced in Table 1 of our paper, our method consistently outperforms RanPAC [1] by a significant margin.
> >
> > Q2: Furthermore, if the matrix is learnable, I am not confident about (or at least it is not clarified or analyzed) what kind of correlation can be learned and how to make sure a valid correlation can be learned, especially considering that the matrix and other parameters are learned simultaneously.
> >
> > A2: Thank you for raising this important point. We would like to clarify that the correlation matrix $\boldsymbol{M}$ in our method is **not a directly learnable entity**. Instead, it is a temporary variable computed based on the learnable features (Eq. 1 of our paper) and is subsequently used to calculate the loss functions $\mathcal{L}\_{\mathrm{diag}}$ and $\mathcal{L}\_{\mathrm{rdn}}$. As such, the correlation matrix itself is not directly learned, but rather it is determined once all model parameters and inputs are set.
> >
> > In fact, the correlation matrix characterizes the **relationship between feature embeddings of PTM and the slow learner**. Concretely, the $i$-th row and $j$-th column of $\boldsymbol{M}$ measures the correlation between the $i$-th feature dimension (also termed as channel or pattern in the literature) of the PTM and the $j$-th feature dimension of the slow learner.
> >
> > Q3: For the ablation studies on aggregation, I want to see how important aggregation inference is and how’s the sensitivity of it, for an already-trained model. It seems Table 2 is the ablation studies of whether slow or fast is included in the whole model, mainly the training process.
> >
> > A3: Thank you for your insightful question.
> >
> > We argue that aggregation is important as it facilitates interaction between the slow and fast learner, aligning with the complementary theory. **The importance of our aggregation** can be demonstrated through the following experiments: 1) As shown in Table 7 of the rebuttal PDF, the proposed aggregation method results in an average improvement of 2.1% in final accuracy across six datasets compared to the baseline. 2) Table 3 of the manuscript further highlights that our aggregation strategy leads to an improvement of at least 0.53% in final accuracy when compared to other competitive aggregation alternatives.
> >
> > Regarding **the sensitivity of aggregation inference**, it's worth noting that the only hyperparameter involved is $\gamma$. One can observe in Table 10 of the rebuttal PDF that $\gamma$ remains relatively stable within a certain range (i.e., $\gamma$ varies from 0.1 to 10). In our method, $\gamma$ is simply set to 1, as stated in line 246 of the manuscript.

---

> > ### Author Response · Authors · 2024-08-13
> > **Further Clarifications (2/2)**
> >
> > Q4: The authors provide a new analysis on the aggregation weights in rebuttal - “in combination with the results in Fig 4, FL and SL primarily exhibits higher weights in sessions 7-10 and 1-6 respectively”. How to explain this observation?
> >
> > A4: Thank you for your question.
> > We would like to provide **further clarification regarding Figure 7** in the rebuttal PDF. Figure 7 depicts the average aggregation weights of both learners after the whole model is trained in the final incremental session. The horizontal axis represents the session number to which each test sample belongs. For instance, "1" in the horizontal axis indicates test samples belong to classes in session 1 (e.g., classes 0-19 in ImageNet-R). The vertical axis shows the average aggregation weights of each learner assigned to these test samples.
> >
> > Thus, the results presented in Figure 7, in conjunction with those in Figure 4, are intended to illustrate how **the aggregated model dynamically leverages the strengths of both learners**. For example, in sessions 7-10 (classes 120-199), the fast learner consistently shows higher weights, which is consistent with its superior classification accuracy in these classes as depicted in Figure 4. This highlights the fast learner's adaptability. Conversely, in sessions 1-6 (classes 0-119) of Figure 7, the slow learner obtains higher weights, generally aligning with its demonstrated stability and better performance in these classes shown in Figure 4. By adaptively balancing the contributions of both learners, our method achieves a harmonious trade-off between stability and adaptability. This dynamic aggregation is key to the state-of-the-art performance we report across six datasets.
> >
> > For the readers to have a better understanding, we will replace the session numbers in Figure 7 in our revised paper with the corresponding class numbers which directly correspond to the horizontal axis in Figure 4.
> >
> > Q5: About the limitation - my comments are only about discussing more essential limitations in the paper.
> >
> > A5: We appreciate your concern regarding the limitations of our method. Our approach is indeed built upon RanPAC [1], and as such, it shares some of the same limitations. For instance, our method **relies on a strong feature extractor** to effectively inherit generalizability from PTMs, making it less suitable for scenarios where training needs to be performed from scratch or starting from rather small tasks. However, existing continual learning methods that utilize self-supervised learning or otherwise create good feature extractor backbones could potentially draw inspiration from our method for downstream continual learning tasks.
> >
> > Additionally, our method introduces three hyper-parameters to balance the loss functions during training, as previously discussed. While our experiments demonstrate that a set of default values works well across the six datasets included in the manuscript, we acknowledge that **these choices might not be optimal** when applied to datasets with essentially different statistical characteristics. Furthermore, the search for these hyperparameters currently lacks a strong theoretical foundation, which is an area that could benefit from further research and refinement.
> >
> > To date, these are the primary limitations we have identified. We believe that acknowledging and addressing these limitations will help in understanding the scope and applicability of our method.
> >
> > We sincerely hope that this response addresses your concerns, and we remain fully open to any further discussion to ensure the highest quality and clarity of our work.
> >
> > Thank you again for your time and consideration.
> >
> > [1] RanPAC: Random Projections and Pre-trained Models for Continual Learning. NeurIPS2023.

---

### Author Rebuttal · Authors · 2024-08-07

We sincerely appreciate the valuable comments from all the reviewers. We diligently provide detailed explanations for the questions raised in the respective comments section **point-to-point**.

In addition, **supplementary experiments and theoretical analyses** are incorporated into the one-page PDF attached to the global response. Specifically, the PDF encompasses the following:

1. **Detailed hyper-parameter sensitivity analyses.** This supports our claim in the paper that a set of default choices can be used, and our method does not require exhaustive hyper-parameter tuning.
2. **A Broad study of each proposed component over six datasets.** This validates the necessity and effectiveness of each module in our framework.
3. **Theoretical analyses of our method** using the information bottleneck principle which demonstrates the interpretability of the proposed approach.
4. **Additional validation on DomainNet** which verifies our method is robust to data distribution changes in continual learning.
5. Other comparison and ablation experiments suggested by the reviewers.

We will include these experiments and explanations in the revised version of our paper as well as its supplement, for the readers to have a better understanding.

---

### Decision · Program_Chairs · 2024-09-25

**Decision:**

Accept (poster)

**Comment:**

This paper focuses on continual learning. There is a growing interest in using pre-trained models instead of starting from scratch in these problems. Rather than using PEFT, which can limit the exploration of pre-trained models, it reduces the model’s adaptability to new concepts. To address these issues, the authors propose a new framework inspired by slow-fast learning, which includes a transfer loss function to inherit general knowledge from pre-trained models and uses slow-tuning parameters to capture more informative features. To balance stability and adaptability, the framework fixes the slow tuning parameters while continuously updating the fast ones, incorporating novel concepts and preventing catastrophic forgetting. An entropy-based aggregation strategy is introduced during inference to leverage the complementarity between slow and fast learners. Extensive experiments on six benchmark datasets show that this method outperforms its counterparts.

Although there have been some negative comments on the paper, after carefully reviewing both the paper and the discussions between the authors and reviewers, I support accepting it. I believe the idea presented is valuable to the community. I encourage the authors to address the feedback from the reviews and make the necessary revisions to strengthen the paper.